# MHC class I and MHC class II reporter mice enable analysis of immune oligodendroglia in mouse models of multiple sclerosis

Em P Harrington[1,2,3]*[†], Riley B Catenacci[1,2][†], Matthew D Smith[1], Dongeun Heo[2], Cecilia E Miller[1], Keya R Meyers[1], Jenna Glatzer[2], Dwight E Bergles[2,4], Peter A Calabresi[1,2]

[1]Department of Neurology, Johns Hopkins University School of Medicine, Baltimore, United States; [2]The Solomon H. Snyder Department of Neuroscience, Johns Hopkins University School of Medicine, Baltimore, United States; [3]Department of Neurology, The Ohio State University College of Medicine, Columbus, United States; [4]The Kavli Neuroscience Discovery Institute, Johns Hopkins University, Baltimore, United States

**Abstract** Oligodendrocytes and their progenitors upregulate MHC pathways in response to inflammation, but the frequency of this phenotypic change is unknown and the features of these immune oligodendroglia are poorly defined. We generated MHC class I and II transgenic reporter mice to define their dynamics in response to inflammatory demyelination, providing a means to monitor MHC activation in diverse cell types in living mice and define their roles in aging, injury, and disease.

**\*For correspondence:**
emily.harrington@osumc.edu

[†]These authors contributed equally to this work

**Competing interest:** The authors declare that no competing interests exist.

## Editor's evaluation

This study reports an important new resource, MHC class I and MHC class II reporter mice, which provide a means to monitor MHC activation in vivo. The authors use these mice to study inflammatory demyelination in two mouse models of multiple sclerosis. The study provides a compelling demonstration of the new reporter lines as valuable tools for the analysis of inflammation and neurodegeneration.

## Introduction

Single-cell and single-nucleus RNA sequencing has revealed that some oligodendroglia in both mouse inflammatory models (*Falcão et al., 2018*; *Meijer et al., 2022*) and human multiple sclerosis (MS) (*Schirmer et al., 2019*; *Jäkel et al., 2019*; *Absinta et al., 2021*) express transcripts associated with major histocompatibility complex (MHC) antigen presenting and processing pathways. These immune oligodendrocyte precursor cells (iOPCs) and oligodendrocytes (iOLs) have been detected in Alzheimer's disease (*Lau et al., 2020*) and viral infection models (*Pan et al., 2020*; *Malone et al., 2008*; *Phares et al., 2009*), and can be induced by exposure to interferon-γ (IFN-γ), suggesting that some oligodendroglia undergo this distinct phenotypic change in response to inflammation. The role of these immune oligodendroglia is unknown (*Harrington et al., 2020*), but their presence raises the possibility that oligodendroglia may present antigens to T cells, be subject to cytotoxic CD8 T cell-mediated death (*Kirby et al., 2019*) and perpetuate the immune response through release of cytokines and interactions with CD4 T cells. This oligodendroglial death or inflammation could contribute

**eLife digest** Nerve cells in the brain and spinal cord are surrounded by a layer of insulation called myelin that allows cells to transmit messages to each other more quickly and efficiently. This protective sheath is produced by cells called oligodendrocytes which together with their immature counterparts can also repair damage caused to myelin.

In the inflammatory disease multiple sclerosis (MS), this insulation is disrupted and oligodendroglia fail to repair breaks in the myelin sheath, leaving nerves vulnerable to further damage. Recently it was discovered that mature and immature oligodendrocytes (which are collectively known as oligodendroglia) sometimes express proteins normally restricted to the immune system called major histocompatibility complexes (or MHCs for short). Researchers believe that MHC expression may allow oligodendroglia to interact with immune cells, potentially leading to the removal of oligodendroglia by the immune system as well as inflammation that exacerbates damage to nerves and hinders myelin repair.

Knowing when oligodendroglia start producing MHCs and where these MHC-expressing cells are located is therefore important for understanding their role in MS. However, it is difficult to identify the location of MHC-expressing oligodendroglia using methods that are currently available. To address this, Harrington, Catenacci et al. created a genetically engineered mouse model in which the MHC-expressing oligodendroglia also generated a red fluorescent protein that could be detected under a microscope. This revealed that only a small number of oligodendroglia in the nervous system had MHCs, but these cells were located in areas of the brain and spinal cord with the highest inflammatory activity.

Further microscopy studies in mice that developed MS-like symptoms revealed that MHC production in oligodendroglia increased compared with healthy animals, and that the proportion of oligodendroglia that produced MHC was highest in mice with the most severe symptoms. MHC-expressing oligodendroglia also congregated in the most damaged areas of the brain and spinal cord.

These results suggest that MHC expression may contribute to inflammation and impact the function of oligodendroglia that have these molecules. In the future, Harrington et al. hope that their new mouse model will help researchers study the role of MHC expression in different diseases, and in the case of MS, aid the development of new treatments.

to impaired remyelination seen in MS and other progressive diseases (*Lubetzki et al., 2020*; *Mahad et al., 2015*; *Lassmann et al., 2012*). Further exploration of the spatial and temporal dynamics of iOPCs/iOLs have been limited by their relative rarity and our inability to identify which cells have transformed in living tissue. To enable detection of which cells upregulate MHC pathways in vivo, we generated two novel MHC I and MHC II reporter mouse lines that express tdTomato when these pathways are activated, and used these lines to define their incidence and transcriptional characteristics in two mouse models of inflammatory demyelination.

## Results

### MHC reporter mice are a reliable readout of MHC protein expression

MHC class II chaperone invariant chain (Cd74 or Ii) and MHC class I component beta-2-microglobulin (B2m) are required components of the antigen processing/presentation machinery that are expressed by a subset of oligodendroglia in mouse inflammatory models (*Falcão et al., 2018*) and human MS brain (*Schirmer et al., 2019*; *Jäkel et al., 2019*). We chose to target *B2m* and *Cd74* genes for the generation of MHC class I and MHC class II reporter mice, respectively, as both of these transcripts are upregulated in oligodendroglia in response to IFN-γ treatment (*Kirby et al., 2019*), and both transcripts are present in inflammatory oligodendroglia in mouse inflammatory models (*Falcão et al., 2018*) and human MS tissue (*Schirmer et al., 2019*; *Jäkel et al., 2019*). To facilitate identification of these cells, we used CRISPR/Cas9-mediated gene editing to replace the stop codon of these genes with a P2A-TdTomato-WPRE-pA sequence (*Figure 1A-C*), generating *B2m-tdTomato* (*B2m^{tdT}*) and *Cd74-tdTomato* (*Cd74^{tdT}*) reporter mice in which 2A cleaving self-peptide facilitates cleavage

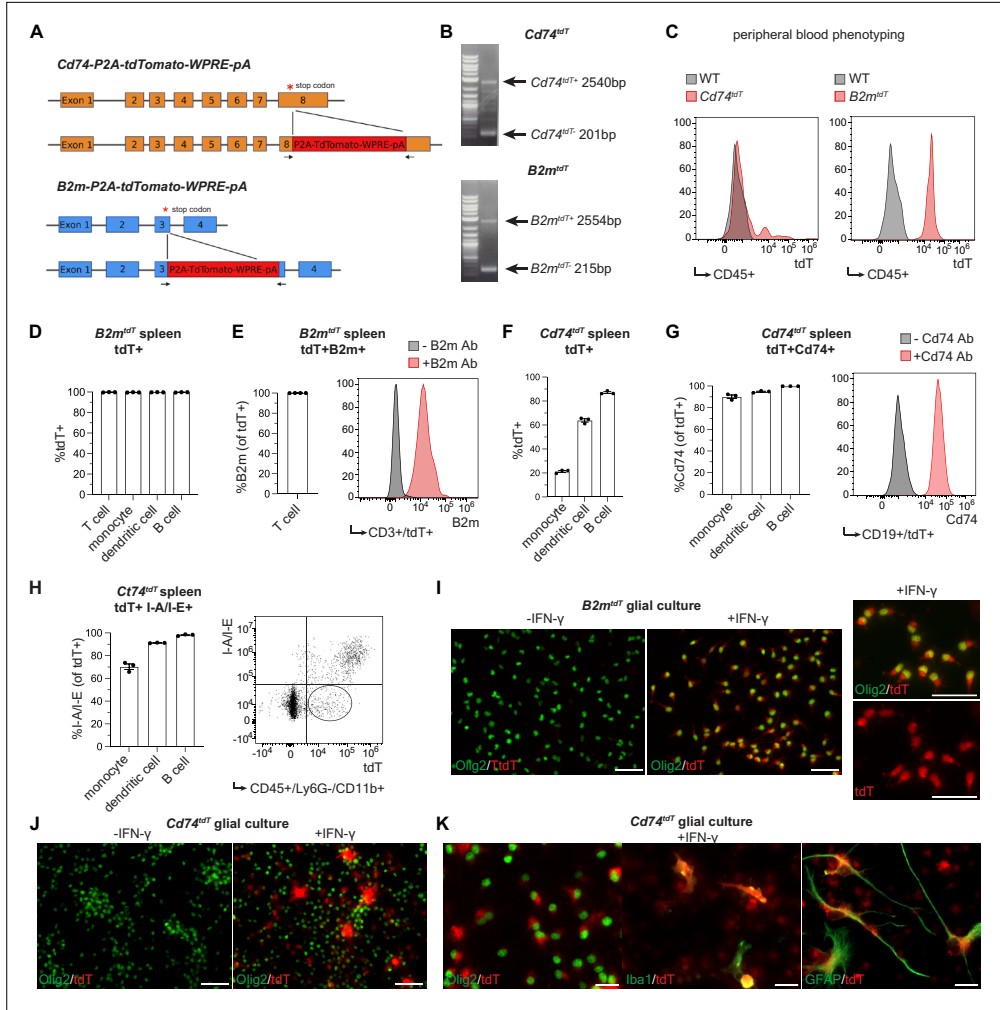

**Figure 1.** Major histocompatibility complex (MHC) reporter mice validation. (**A**) Stop codons of *Cd74* and *B2m* are replaced by *P2A-TdTomato-WPRE-pA* construct to generate *Cd74^tdT^* and *B2m^tdT^* reporter mice, respectively. Figure made in https://www.biorender.com/. (**B**) Genotyping of reporter mice with primers (indicated by arrows in panel A) spanning reporter construct insertion site. (**C**) Representative histogram of flow cytometry of endogenous tdT expression in peripheral blood CD45+ cells for phenotyping reporter and wild-type mice. (**D**) Quantification of flow cytometry percentage of endogenous tdT expression in *B2m^tdT^* splenic cell populations. n=3 mice. (**E**) Quantification of flow cytometry percentage of B2m co-expression in tdT-positive CD3+ T cells in *B2m^tdT^* spleen with representative histogram of B2m expression with and without B2m antibody. n=3 mice. (**F**) Quantification of flow cytometry percentage of endogenous tdT expression in *Cd74^tdT^* splenic cell populations. n=3 mice. (**G**) Quantification of flow cytometry percentage of Cd74 co-expression in tdT-positive splenic cell populations in *Cd74^tdT^* spleen with representative histogram of tdT-positive CD19+ B cells stained with and without CD74. n=3 mice. (**H**) Quantification of flow cytometry percentage of I/A-I/E co-expression with tdT-positive splenic cell populations in *Cd74^tdT^* spleen with representative histogram of tdT and I-A/I-E expression in monocytes with tdT-intermediate I-A/I-E-negative population outlined. n=3 mice. (**I**) Immunopanned *B2m^tdT^* post-natal pup glial culture with representative images of tdT expression in Olig2+ oligodendroglia with and without interferon-γ (IFN-γ) treatment. Scale bars, 50 µm. (**J**) Immunopanned *Cd74^tdT^* post-natal pup glial culture with representative images of tdT expression in Olig2+ oligodendroglia with and without IFN-γ treatment. Scale bars, 50 µm. (**K**) Representative images of tdT expression in Olig2+ oligodendroglia, GFAP+ astrocytes, and Iba1+ microglia in *Cd74^tdT^* glial culture with IFN-γ treatment. Scale bars, 20 µm. Data represented are means ± s.e.m.

The online version of this article includes the following source data and figure supplement(s) for figure 1:

**Source data 1.** Generation of *Cd74^tdT^* and *B2m^tdT^* reporter animals.

**Source data 2.** Data from analysis of flow cytometry, sheets are labeled with letter corresponding to data panels in *Figure 1*.

*Figure 1 continued on next page*

*Figure 1 continued*

**Source data 3.** Representative gating strategy from flow cytometry depicted in *Figure 1*.

**Source data 4.** Original gel image files of PCR genotyping for transgenic reporter founder animals for Figure 1B.

**Figure supplement 1.** Western blots of *B2m*[tdT] and *Cd74*[tdT] reporter and wild-type animals.

**Figure supplement 1—source data 1.** Original western blot images.

**Figure supplement 1—source data 2.** Data from analysis of western blot band intensities.

**Figure supplement 2.** *Cd74*[tdT] splenocyte and *OT-II* CD4 T cell co-culture.

**Figure supplement 2—source data 1.** Data from analysis of flow cytometry.

of tdTomato from the endogenous protein without disrupting expression of the endogenous genes (*Figure 1—figure supplement 1*) or antigen presentation (*Figure 1—figure supplement 2*).

To determine if these transgenes accurately report transcriptional activation of MHC components, we examined splenic immune cells known to express MHC class I and II. In *B2m*[tdT] reporter mice, tdT was ubiquitously expressed in all immune cells analyzed in the spleen (*Figure 1D*) and tdT fluorescent cells were immunoreactive to B2m (*Figure 1E*). In *Cd74*[tdT] reporter mice, tdT expression was highest in professional antigen presenting cells, such as dendritic cells and B cells (*Figure 1F*) and tdT fluorescent cells were immunoreactive to CD74 (*Figure 1G*). Independent of MHC class II, Cd74 has diverse roles in cell survival, migration and MIF (macrophage migration inhibitory factor) cytokine signaling (*Su et al., 2017*; *Schröder, 2016*). Thus, we determined the relationship between tdT and MHC class II expression using I-A/I-E MHC antibodies. I-A/I-E co-expression with tdT was highest in dendritic cells and B cells (*Figure 1H*) and monocytes that exhibited intermediate tdT fluorescence were also negative or weakly immunoreactive for I-A/I-E expression (*Figure 1H*), indicating that the level of tdT expression correlates well with MHC class II receptor expression.

To determine whether reporter expression is induced in glial cells, immunopanned reporter expressing glial cells were cultured with and without IFN-γ, as we have previously shown that IFN-γ is sufficient to induce MHC class I and II expression in oligodendroglia in vitro (*Kirby et al., 2019*). Consistent with this observation, IFN-γ increased tdT expression in Olig2 immunoreactive (+) cells in OPC enriched cultures (*Fancy et al., 2011*) from both *B2m*[tdT] and *Cd74*[tdT] (*Figure 1I–J*). In *Cd74*[tdT] glial

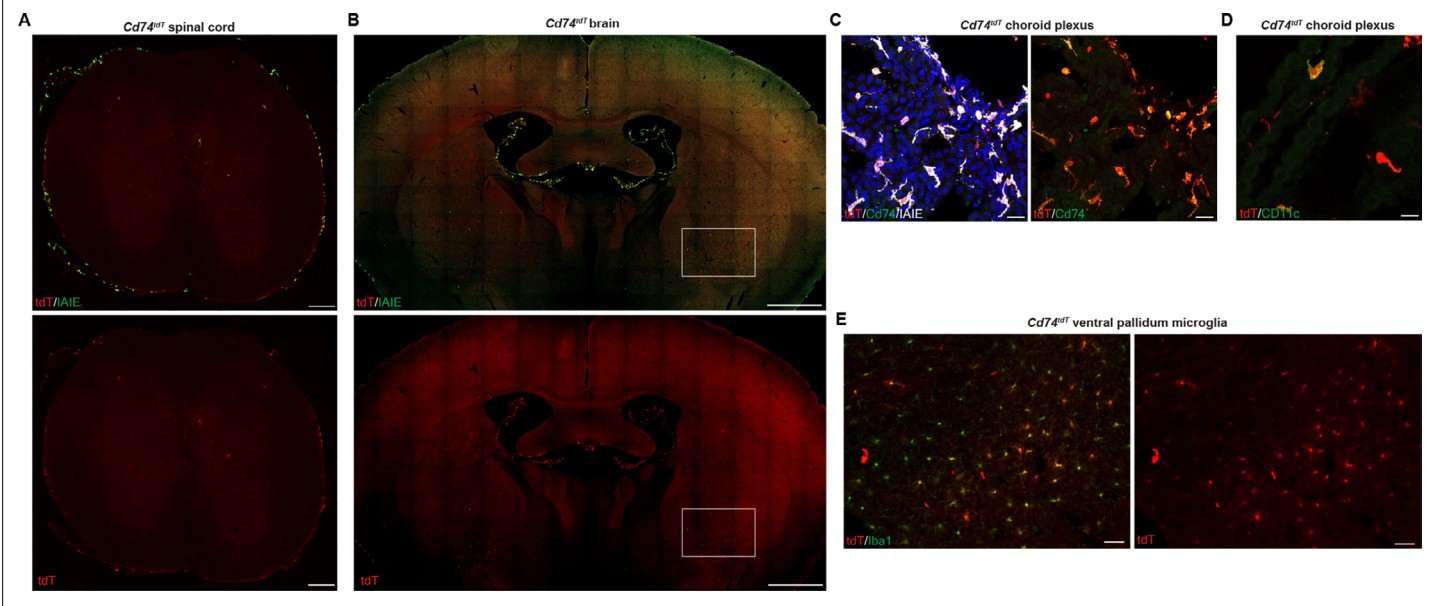

**Figure 2.** *Cd74*[tdT] reporter adult baseline central nervous system (CNS) tdT expression. (**A**) Representative images of *Cd74*[tdT] endogenous tdT reporter expression in adult spinal cord stained for I-A/I-E. Scale bars, 200 µm. (**B**) Representative images of *Cd74*[tdT] endogenous tdT reporter expression in adult brain stained for I-A/I-E. Scale bars, 1 mm. (**C**) Confocal images of adult brain choroid plexus endogenous tdT stained with Cd74 and I-A/I-E antibodies. Scale bars, 20 µm. (**D**) Confocal images of choroid plexus endogenous tdT on CD11c+ dendritic cells. Scale bars, 20 µm. (**E**) tdT expression on subset of Iba1+ microglia in ventral pallidum (magnification of box in **B**). Scale bars, 50 µm. *Cd74*[tdT] adult CNS reporter expression n=6 adult mice.

cultures treated with IFN-γ, tdT expression was also found in Iba1+ microglia and GFAP+ astrocytes (*Figure 1K*).

To characterize the adult central nervous system (CNS) baseline MHC expression, we evaluated the brain, spinal cord, and retinal tissues of adult *Cd74$^{tdT}$* and *B2m$^{tdT}$* reporter mice. In *Cd74$^{tdT}$* mice, tdT expression was most prevalent in the meninges and choroid plexus (*Figure 2A–D*), but was also observed within some microglia in the ventral brain (*Figure 2B and E*). tdT expression co-localized with I-A/I-E (*Figure 2A–C*) and Cd74 (*Figure 2C*) expression in the meninges and choroid plexus. In the CNS of *B2m$^{tdT}$* mice, tdT expression was prominent in endothelial cells and microglia throughout the brain, spinal cord, and retina (*Figure 3A–G*). In addition, some neurons in the cerebellum, septum, hippocampus, and cortex expressed tdT (*Figure 3H–L*). Within the corpus callosum and cortex, in addition to microglial and endothelial cell expression, there was diffuse parenchymal expression of tdT (*Figure 3A–C*), possibly arising from projections of tdT+ neurons. Expression of MHC class I by these neurons may allow them to present antigen and render them more prone to death in inflammatory conditions.

## MHC reporter expression is induced in EAE

To evaluate CNS activation of MHC pathways in distinct cell types under inflammatory conditions, we induced experimental autoimmune encephalitis (EAE) by MOG$_{35-55}$ peptide immunization. Reporter mice subjected to MOG$_{35-55}$ EAE were sacrificed with EAE presentation ranging from pre-clinical (score 0), tail and hindlimb weakness (score 1–2.5) to complete hindlimb paralysis (score 3–4). With increasing EAE clinical score, there was a concomitant increase in tdT expression in spinal cord lesion areas in both *B2m$^{tdT}$* (*Figure 4A*) and *Cd74$^{tdT}$* animals (*Figure 4B*). tdT was also present in EAE ventral brain meningeal cell clusters and corpus callosum cell clusters in *Cd74$^{tdT}$* and *B2m$^{tdT}$* animals (*Figure 5A–F*). tdT was expressed in both infiltrating myeloid cells and microglia in *B2m$^{tdT}$* and *Cd74$^{tdT}$* EAE brain (*Figure 5G*).

When analyzed by flow cytometry, infiltrating myeloid cells and microglia exhibited tdT expression in the brain and spinal cord of *Cd74$^{tdT}$* mice with EAE (*Figure 6A and B*) and both the proportion of myeloid cells and microglia that expressed tdT (*Figure 6A*), and the level of tdT expression by these cells (*Figure 6B*), were higher in co-stimulatory molecule expressing cells compared to non-co-stimulatory molecule expressing cells. In *B2m$^{tdT}$* mouse brain, the percentage of tdT expressing microglia was not significantly different between M1 or M2 microglia, defined by CD86 and CD206 expression, respectively (*Jurga et al., 2020*; *Figure 6C and D*); however, M1 microglia had significantly higher tdT fluorescence compared to M2 and M1-2 microglia (*Figure 5E*). Together, this analysis highlights the ability of these reporter mice to reliably identify endogenous and infiltrating cells that upregulate MHC pathways.

## MHC reporter positive immune oligodendroglia correlate with degree of inflammation

To determine whether oligodendroglia express MHC reporters in the setting of inflammation, MOG$_{35-55}$ immunized brain and spinal cord were analyzed from non-immunized baseline, pre-clinical EAE score 0 (8 days post-immunization [dpi]), and early to peak EAE (8–17 dpi) with a range of EAE clinical scores from 1.5 to 3.5. In both *B2m$^{tdT}$* and *Cd74$^{tdT}$* MOG$_{35-55}$ EAE spinal cord, Olig2+ oligodendroglia were found to express tdT within EAE lesions (*Figure 7A*). To quantify oligodendroglial tdT expression, four spinal cord regions (dorsal horn, lateral white matter, dorsal white matter, and central cord) were quantified for Olig2 and endogenous tdT expression in EAE lumbar spinal cord sections (*Figure 7B and C*). The number of tdT+Olig2+ oligodendroglia was significantly higher in clinically symptomatic compared to pre-symptomatic score 0 animals (*Figure 7D*) in both *B2m$^{tdT}$* (pre: 1.0 ± 0.2%, n=4; post: 6.3 ± 1.8%, n=9, p=0.0056 unpaired Mann-Whitney t-test) and *Cd74$^{tdT}$* spinal cord (pre: 0.02 ± 0.01%, n=3; post: 0.6 ± 0.2%, n=10, p=0.0070 unpaired Mann-Whitney t-test).

tdT+ oligodendroglia were more abundant in EAE lesion areas, determined by DAPI hypercellularity, compared to non-lesion areas in *B2m$^{tdT}$* EAE mice (p=0.0156 paired Wilcoxon t-test, n=8) but not *Cd74$^{tdT}$* EAE mice (p=0.1289 paired Wilcoxon t-test, n=9) (*Figure 7E and F*). To quantify the abundance of tdT+ oligodendroglia in relation to the degree of inflammatory infiltrates, CD45 density was quantified on whole spinal cord sections. An increase in CD45 density was significantly correlated with a higher percentage of tdT+ oligodendroglia in *B2m$^{tdT}$* EAE sections, but not *Cd74$^{tdT}$* EAE sections

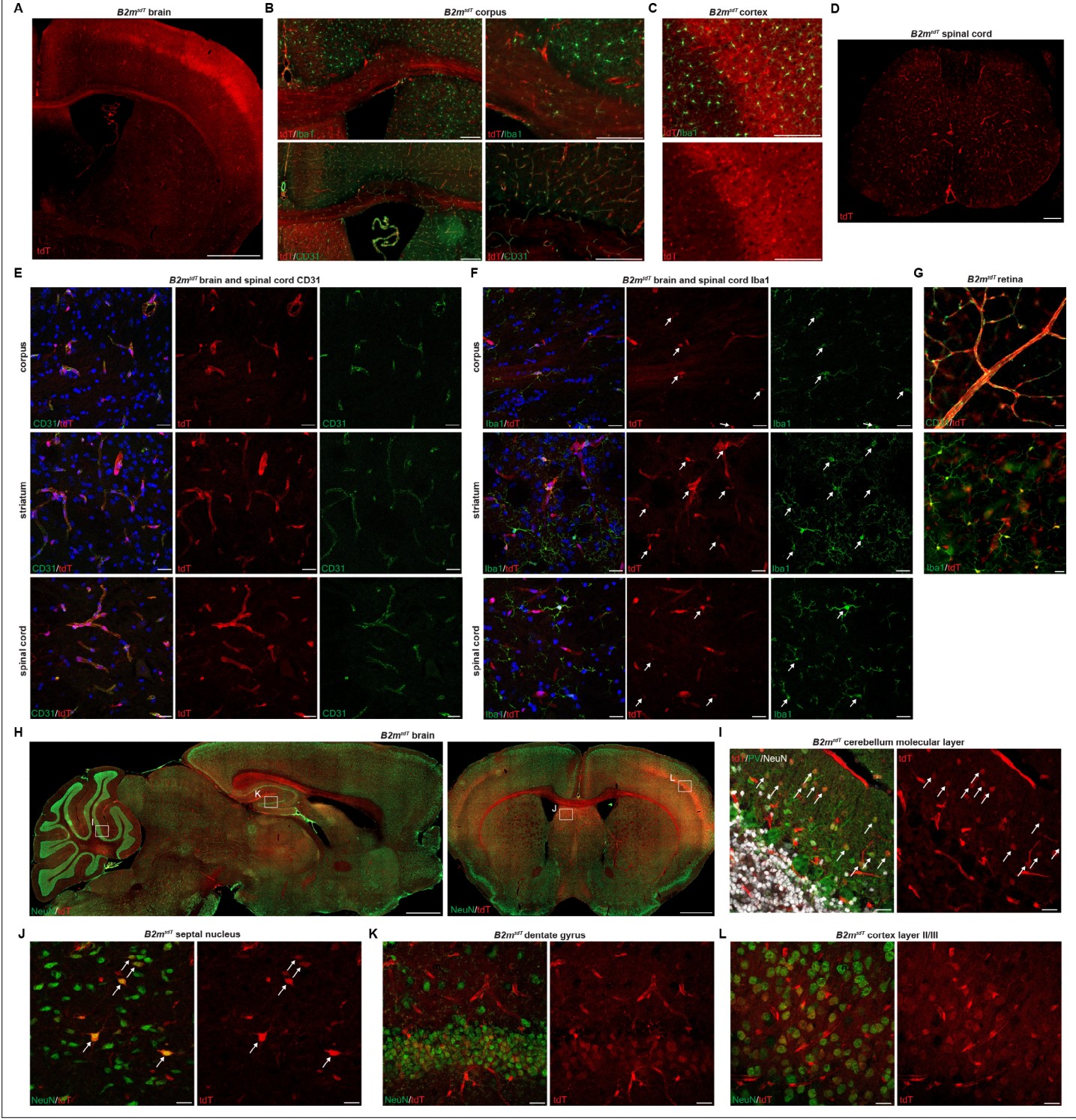

**Figure 3.** $B2m^{tdT}$ reporter baseline central nervous system (CNS) TdT expression. (**A**) Representative images of $B2m^{tdT}$ endogenous tdT reporter adult expression in adult brain. Scale bar, 1 mm. (**B**) Co-localization of tdT in $B2m^{tdT}$ corpus callosum with Iba1+ microglia and CD31+ endothelial cells. Scale bars, 200 µm. (**C**) Co-localization of tdT with Iba1 in $B2m^{tdT}$ cortex. Scale bar, 200 um. (**D**) $B2m^{tdT}$ endogenous tdT reporter expression in adult spinal cord. Scale bar, 200 µm. (**E**) Confocal imaging of corpus callosum, striatum, and spinal cord endogenous tdT expression on CD31+ endothelial cells. Scale bars, 20 µm. (**F**) Confocal imaging of corpus callosum, striatum, and spinal cord endogenous tdT expression on Iba1+ microglia with tdT+Iba1+ cell bodies indicated with arrows. Scale bars, 20 µm. (**G**) Flat mount adult $B2m^{tdT}$ retina with TdT expression on CD31+ endothelial cells and Iba1+ microglia. Scale bars, 20 µm. (**H**) $B2m^{tdT}$ endogenous tdT reporter expression in sagittal brain and coronal brain stained with NeuN (boxes indicate

*Figure 3 continued on next page*

*Figure 3 continued*

regions of interest confocal imaging in panels I–L). Scale bars, 1 mm. (**I**) Confocal imaging of endogenous tdT+ parvalbumin PV+ neurons, indicated by arrows, in cerebellar molecular layer. Scale bars, 20 μm. (**J**) Confocal imaging of septal nucleus with scattered tdT+NeuN+ neurons indicated by arrows. Scale bars, 20 μm. (**K**) Confocal imaging of hippocampal dentate gyrus tdT+NeuN+ neurons. Scale bars, 20 μm. (**L**) Confocal imaging of cortex layer II/III tdT+NeuN+ neurons. Scale bars, 20 μm. *B2m^{tdT}* adult CNS reporter expression n=6 adult mice.

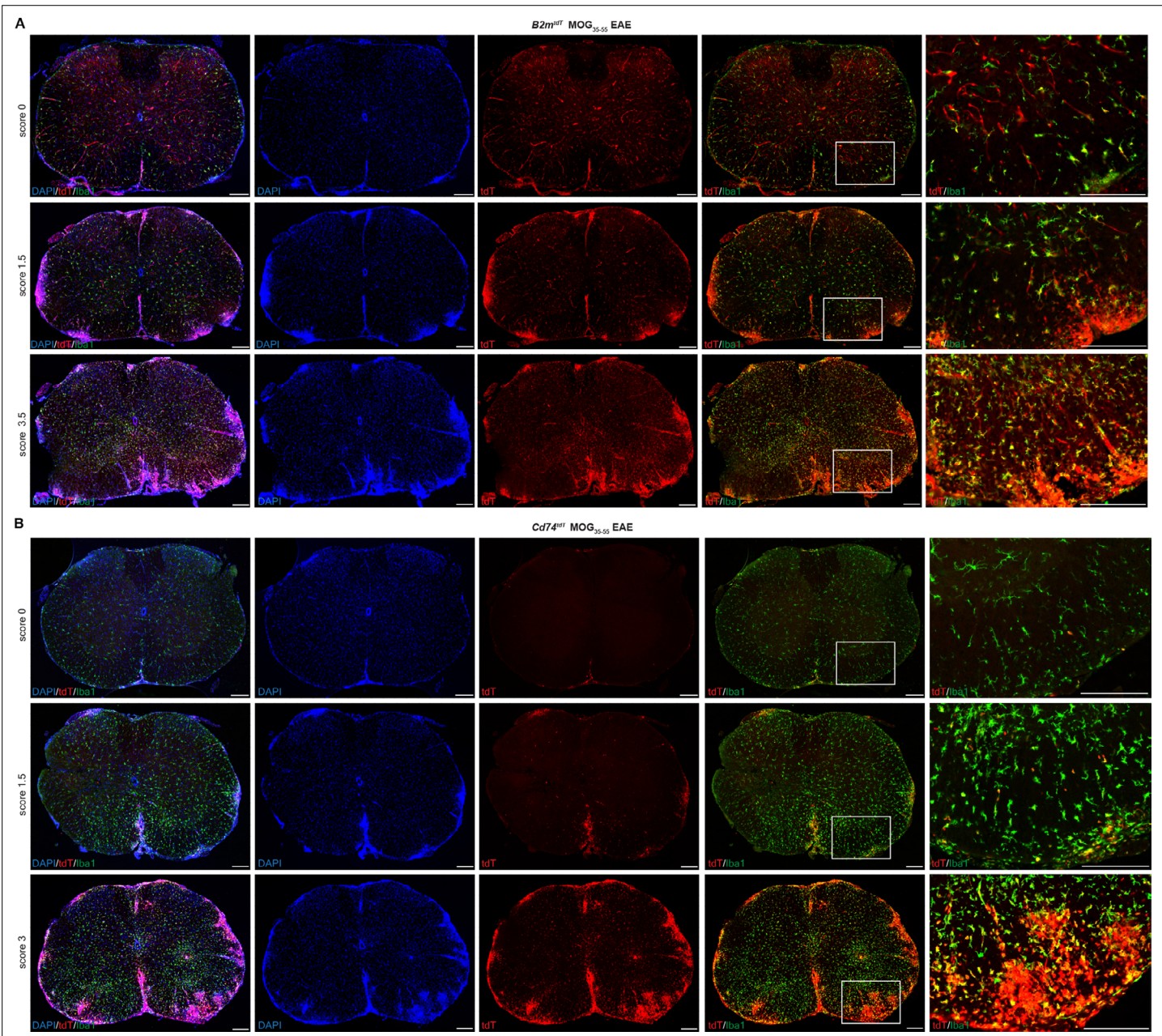

**Figure 4.** *B2m^{tdT}* and *Cd74^{tdT}* reporter expression in MOG$_{35-55}$ peptide immunized experimental autoimmune encephalitis (EAE) spinal cord. (**A**) *B2m^{tdT}* reporter MOG$_{35-55}$ EAE spinal cord stained with Iba1. tdT expression is present in the vasculature and Iba1+ microglia in pre-clinical score 0 animals and more notable parenchymal expression within EAE lesions indicated by DAPI hypercellularity and Iba1 cell clusters. Scale bars, 200 μm. (**B**) *Cd74^{tdT}* reporter MOG$_{35-55}$ EAE spinal cord stained with Iba1. tdT expression is restricted to the meninges in pre-clinical score 0 mice and with increasing EAE clinical score is prominent within EAE lesions. Scale bars, 200 μm.

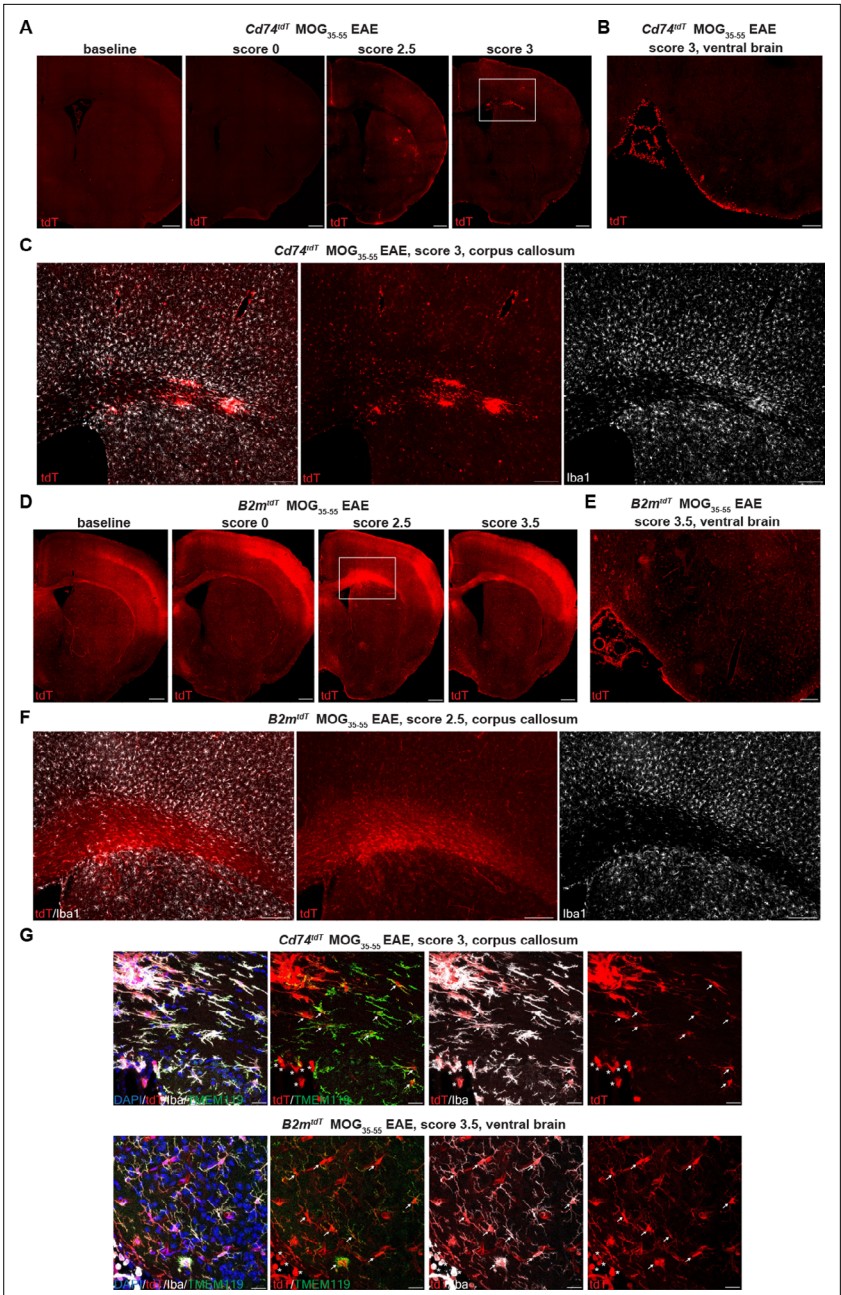

**Figure 5.** *B2m^tdT* and *Cd74^tdT* reporter expression in MOG$_{35-55}$ peptide immunized experimental autoimmune encephalitis (EAE) brain. (**A**) Representative images of *Cd74^tdT* MOG$_{35-55}$ EAE brain with clinical scores >2 demonstrating parenchymal and meningeal tdT expression. Scale bars, 500 µm. (**B**) *Cd74^tdT* EAE ventral brain meningeal cluster of tdT-positive cells. Scale bar, 200 µm. (**C**) *Cd74^tdT* EAE corpus callosum with notable tdT+ infiltrate and tdT parenchymal expression. (**D**) Representative images of *B2m^tdT* MOG$_{35-55}$ EAE brain with meningeal tdT expression and corpus callosum tdT expression. Scale bars, 500 µm. (**E**) *B2m^tdT* EAE ventral brain meningeal cluster of tdT-positive cells. Scale bar, 200 µm. (**F**) *B2m^tdT* EAE corpus callosum with notable tdT+ infiltrate. (**G**) *Cd74^tdT* and *B2m^tdT* EAE brain with tdT+ microglia (Iba1+TMEM119+ indicated by arrows) and tdT+ infiltrating myeloid cells (Iba1+TMEM119- indicated by asterisks). Scale bars 20 µm.

(*Figure 7G*), suggesting higher concentrations of MHC class I oligodendroglia in regions with higher inflammatory activity.

As an alternative measure of inflammatory activity in individual animals, the mean tdT fluorescent intensity of all spinal cord regions of interest (ROIs) imaged was used as a surrogate of overall

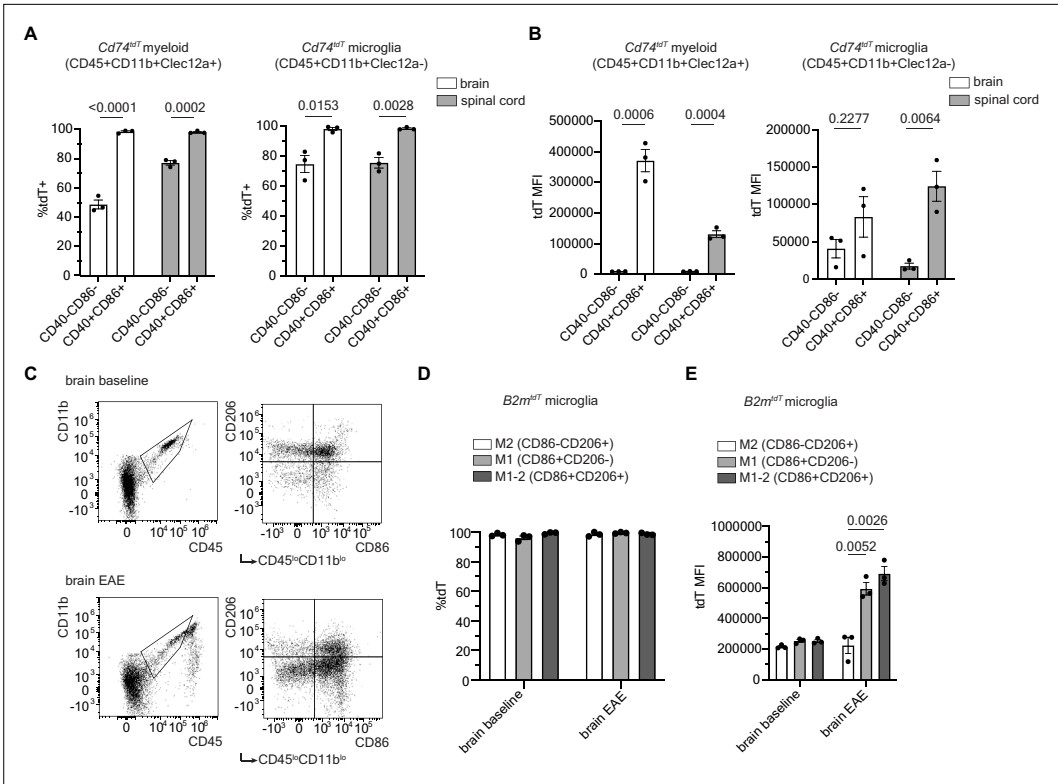

**Figure 6.** Co-stimulatory molecule expression in myeloid and microglial cells in MOG$_{35-55}$ peak experimental autoimmune encephalitis (EAE). (**A**) Quantification of percentage of tdT expression in co-stimulatory molecule positive (CD40+CD86+) compared to co-stimulatory molecule negative (CD40-CD86-) infiltrating myeloid cells (CD45+CD11b+Clec12a+) and microglia (CD45+CD11b+Clec12a-) in *Cd74$^{tdT}$* MOG$_{35-55}$ EAE by flow cytometry. n=3 mice/group. Unpaired t-test. Data represented are means ± s.e.m. (**B**) Quantification of tdT mean fluorescence intensity (MFI) in tdT+ co-stimulatory positive and negative infiltrating myeloid cells and microglia in *Cd74$^{tdT}$* MOG$_{35-55}$ EAE by flow cytometry. n=3 mice/group. Unpaired t-test. Data represented are means ± s.e.m. (**C**) Representative gating strategy for M2 (CD86-CD206+), M1 (CD86+CD206-), and M1-2 microglia (CD86+CD206+). (**D**) Quantification of percentage of tdT expression in M1, M2, and M1-2 microglia in *B2m$^{tdT}$* baseline and EAE brain by flow cytometry. n=3 mice/group. Data represented are means ± s.e.m. (**E**) Quantification of tdT MFI of M1, M2, and M1-2 microglia in *B2m$^{tdT}$* baseline and EAE brain. n=3, mice/group. Unpaired t-test. Data represented are means ± s.e.m.

The online version of this article includes the following source data for figure 6:

**Source data 1.** Data from analysis of flow cytometry, sheets are labeled with letter corresponding to data panels in *Figure 6*.

**Source data 2.** Representative gating strategy from flow cytometry depicted in *Figure 6*.

inflammatory activity, as tdT is expressed on inflammatory infiltrates in both *B2m$^{tdT}$* and *Cd74$^{tdT}$* reporter animals. The individual animal mean ROI tdT intensity was positively correlated with the individual animal mean percentage of tdT+ oligodendroglia in both *B2m$^{tdT}$* and *Cd74$^{tdT}$* reporter mice (*Figure 7H*), highlighting the close correspondence between activation of MHC pathways in oligodendroglia and level of inflammatory activity. Individual animal EAE clinical score was also positively correlated with the mean percentage of tdT+ oligodendroglia in *B2m$^{tdT}$* and *Cd74$^{tdT}$* EAE spinal cord (*Figure 7—figure supplement 1A*).

To determine whether tdT+ oligodendroglia were oligodendrocyte progenitors or mature oligodendrocytes, we performed immunostaining for CC1 and PDGFRa, to identify oligodendrocytes and OPCs, respectively. In naive *B2m$^{tdT}$* spinal cords, tdT+ oligodendroglia were predominantly mature CC1+ oligodendroglia (88.6 ± 1.7%, n=4) (*Figure 7I*). As CC1 labels immune cells as well as mature oligodendrocytes, we were only able to analyze OPCs in EAE sections. A minority of spinal cord TdT+Olig2+ oligodendroglia were immunoreactive to PDGFRa (*Figure 7J*) in *B2m$^{tdT}$* naive (6.2 ± 0.2%,

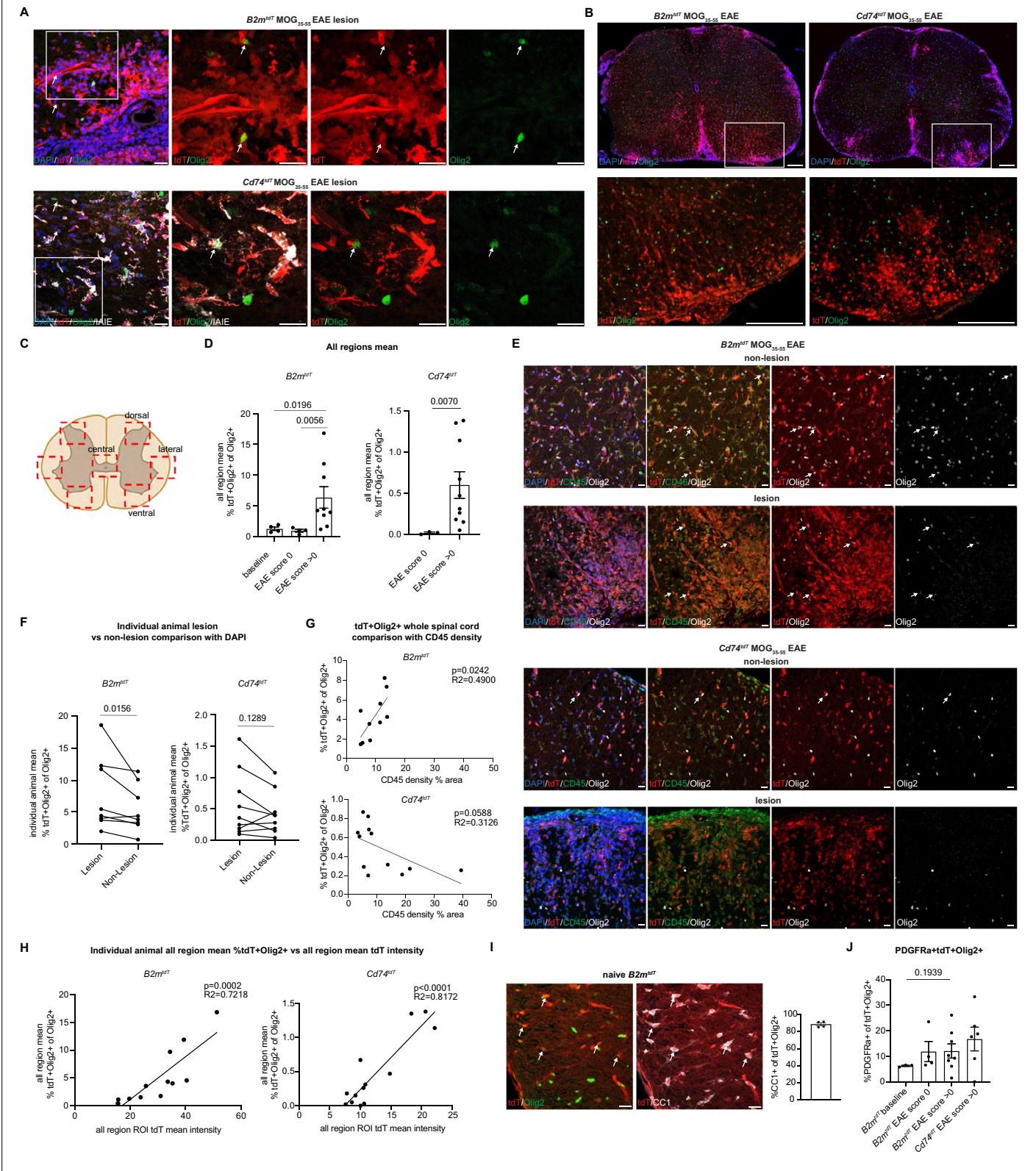

**Figure 7.** Oligodendroglial major histocompatibility complex (MHC) reporter expression in MOG$_{35-55}$ experimental autoimmune encephalitis (EAE) spinal cord. (**A**) Representative confocal imaging of tdT-positive oligodendroglia in MOG$_{35-55}$ EAE *B2m^tdT* and *Cd74^tdT* spinal cord lesions. Endogenous tdT-positive oligodendroglia indicated by arrows. Scale bars, 20 µm. (**B**) Representative images of *B2m^tdT* and *Cd74^tdT* MOG$_{35-55}$ spinal cord whole cord and outlined region of interest (ROI) used for analysis of tdT+ oligodendroglia. Scale bars, 200 µm. (**C**) Diagram of seven ROIs imaged in lumbar

*Figure 7 continued on next page*

Figure 7 continued

spinal cord sections from baseline and EAE spinal cord. Figure made in https://www.biorender.com/. (D) Quantification of percentage of tdT+Olig2+ oligodendroglia in *B2m*$^{tdT}$ and *Cd74*$^{tdT}$ spinal cord. Data points are an individual animal mean across all regions analyzed. n=3–4 mice baseline and score 0, n=9–10 EAE score >0. Unpaired Mann-Whitney t-test. Data represented are means ± s.e.m. (E) Representative confocal imaging of lesion and non-lesion ROIs from the same reporter EAE animal with tdT+Olig2+ oligodendroglia indicated by arrows. Scale bars, 20 μm. (F) Quantification of percentage of tdT+Olig2+ oligodendroglia in lesion compared to non-lesion areas in MOG$_{35-55}$ EAE spinal cord. Data points are an individual animal mean across ROIs with lesions determined by DAPI hypercellularity compared to ROIs without lesions. n=8–9 mice/reporter, paired Wilcoxon t-test. (G) Linear regression of CD45 density on whole spinal cord section compared to percentage of tdT+Olig2+ oligodendroglia in whole spinal cord section in MOG$_{35-55}$ EAE spinal cord. Data points are an individual spinal cord sections. n=3–4 sections/mouse, 3 mice/reporter. (H) Linear regression of all region mean percentage of tdT+Olig2+ oligodendroglia compared to all region mean tdT mean fluorescent intensity in MOG$_{35-55}$ EAE spinal cord. n=13 mice/reporter. Data points are individual animals. (I) Representative confocal imaging of adult *B2m*$^{tdT}$ spinal cord at baseline with endogenous tdT co-localization with Olig2+CC1+ mature oligodendroglia (arrows). Quantification of percentage of CC1+tdT+ Olig2+ oligodendrocytes in *B2m*$^{tdT}$ baseline adult spinal cord. n=4 mice. Scale bars, 20 μm. Data represented are means ± s.e.m. (J) Quantification of percentage of PDGFRa+tdT+ oligodendroglia in MOG$_{35-55}$ EAE spinal cord. Data points are an individual animal mean across all regions. n=3–4 mice baseline and score 0, n=6–8 mice EAE score >0. Unpaired Mann-Whitney t-test. Data represented are means ± s.e.m.

The online version of this article includes the following source data and figure supplement(s) for figure 7:

**Source data 1.** Immunohistochemistry quantification, sheets are labeled with letter corresponding to data panels in *Figure 7*.

**Figure supplement 1.** Correlation of experimental autoimmune encephalitis (EAE) score with tdT+ oligodendroglia and PDGFRa+tdT+ oligodendroglia in EAE spinal cord.

**Figure supplement 1—source data 1.** Immunohistochemistry quantification, sheets are labeled with letter corresponding to data panels in *Figure 7— figure supplement 1*.

**Figure supplement 2.** Major histocompatibility complex (MHC) reporter expression in astrocytes in MOG35-55 experimental autoimmune encephalitis (EAE) spinal cord lesions.

n=3), *B2m*$^{tdT}$ EAE score 0 (11.8 ± 4.0%, n=4), *B2m*$^{tdT}$ EAE score >0 (12.1 ± 2.8%, n=8), and *Cd74*$^{tdT}$ EAE score >0 (16.8 ± 4.6%, n=6), indicating that most of these MHC expressing oligodendroglia had advanced beyond the progenitor stage. The level of inflammatory activity in an individual animal based on EAE clinical score (*Figure 7—figure supplement 1B*), mean tdT ROI intensity (*Figure 7— figure supplement 1C*), or mean tdT+ oligodendroglia (*Figure 7—figure supplement 1D*) were not significantly correlated with the percentage of tdT+PDGFRa+ immune OPCs, suggesting that immune OPCs are not depleted in setting of more robust inflammatory activity.

To determine if astrocytes demonstrate expression of MHC reporters in the setting of inflammation, EAE spinal cord sections were immunostained for GFAP and Sox9. MHC reporter expression was found in some Sox9+ astrocytes in EAE lesions of both *B2m*$^{tdT}$ and *Cd74*$^{tdT}$ animals (*Figure 7—figure supplement 2*), suggesting that astrocytes also upregulate MHC in the context of inflammation.

In addition to spinal cord tissue, we observed tdT+ oligodendroglia in EAE brains (*Figure 8A and B* and *Figure 8—video 1*). To quantify tdT+ oligodendroglia in EAE brain regions, four regions (ventral, midline corpus, corpus horn, and subventricular zone) were imaged and quantified for Olig2 expression and endogenous tdT expression (without immunostaining) (*Figure 8C and D*). The number of tdT+ oligodendroglia across all brain regions analyzed was significantly higher in clinical scoring EAE animals compared to pre-clinical score 0 animals in both B2m$^{tdT}$ (pre: 1.7 ± 0.7%, n=5; post: 6.1 ± 1.2%, n=9, p=0.0180 unpaired Mann-Whitney t-test) and Cd74$^{tdT}$ mice (pre: ND, n=3; post: 0.5 ± 0.2%, n=11, p=0.0165 unpaired Mann-Whitney t-test) animals (*Figure 8E*). Similar to the spinal cord analysis, the individual animal mean ROI tdT intensity was also positively correlated with the individual animal mean percentage of tdT+ oligodendroglia in the brain of B2m$^{tdT}$ and Cd74$^{tdT}$ EAE mice (*Figure 8F*). However, the EAE clinical score was correlated with the number of TdT+ oligodendroglia in B2m$^{tdT}$ brain, but not the Cd74$^{tdT}$ EAE brain (*Figure 8—figure supplement 1A*).

To determine whether these MHC reporter mice are versatile enough to use with other inflammatory models, we adoptively transferred MOG-reactive Th17 T cells after cuprizone-mediated demyelination (AT-CUP) (*Figure 8G*), which results in inflammatory infiltrates and impaired remyelination in the corpus callosum (*Baxi et al., 2015*). tdT+ oligodendroglia were significantly more abundant in AT-CUP mice across all brain regions analyzed compared to cuprizone alone (CUP) and adoptive transfer (AT) alone in both *B2m*$^{tdT}$ (AT-CUP: 17.4 ± 4.6%, n=6; AT: 2.0 ± 0.6%, n=4; CUP: 0.5 ± 4.6%, n=3) and *Cd74*$^{tdT}$ (AT-CUP: 0.9 ± 0.4%, n=8; AT: 0.1 ± 0.04%, n=6; CUP: ND, n=3) reporter animals (*Figure 8H*). Given that there is a high degree of inflammatory activity in the demyelinated corpus

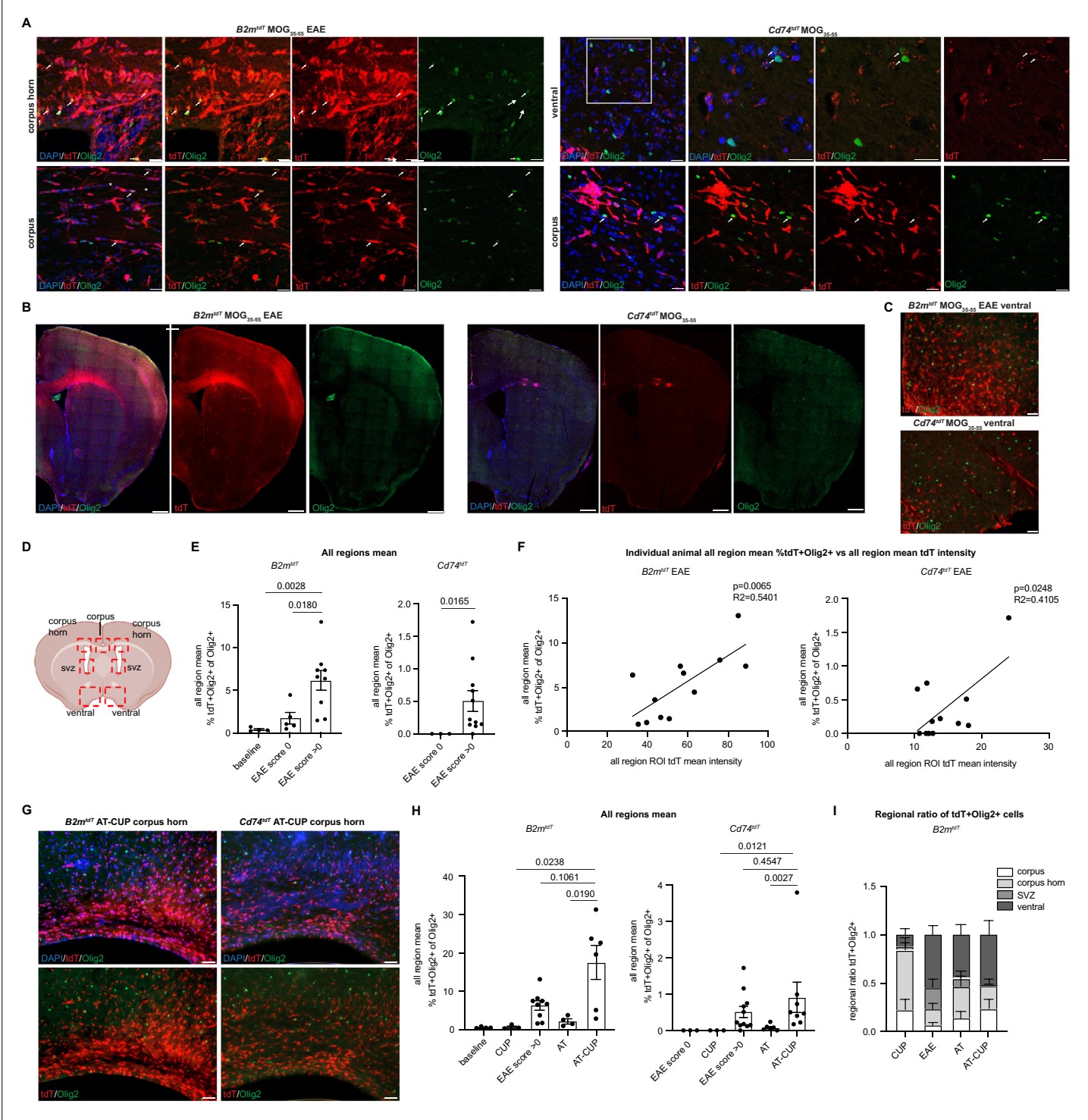

**Figure 8.** Oligodendroglial major histocompatibility complex (MHC) reporter expression in the brain of central nervous system (CNS) inflammatory models. (**A**) Confocal imaging of tdT reporter-positive oligodendroglia in MOG$_{35-55}$ experimental autoimmune encephalitis (EAE) $B2m^{tdT}$ and $Cd74^{tdT}$ brain. Endogenous tdT reporter-positive oligodendroglia indicated by arrows. Scale bars, 20 µm. (**B**) Representative images of MOG$_{35-55}$ EAE $B2m^{tdT}$ and $Cd74^{tdT}$ brain with corpus callosum TdT-positive infiltrate. Scale bars, 500 µm. (**C**) Representative images of $B2m^{tdT}$ and $Cd74^{tdT}$ MOG$_{35-55}$ ventral brain used for region of interest (ROI) analysis of tdT+ oligodendroglia. Scale bars, 50 µm. (**D**) Diagram of seven ROIs imaged in brain sections. Figure made in https://www.biorender.com/. (**E**) Quantification of percentage of tdT+Olig2+ oligodendroglia in $B2m^{tdT}$ and $Cd74^{tdT}$ brain. Data points are an individual animal mean across all regions analyzed. n=3–5 mice baseline and score 0, n=9–11 EAE score >0. Unpaired Mann-Whitney t-test. Data represented are means ± s.e.m. (**F**) Linear regression of all region mean percentage of tdT+Olig2+ oligodendroglia compared to all region mean tdT mean fluorescent

*Figure 8 continued on next page*

*Figure 8 continued*

intensity in MOG$_{35-55}$ EAE brain. n=12 mice/reporter. Data points are individual animals. (**G**) Representative images of *B2m*$^{tdT}$ and *Cd74*$^{tdT}$ AT-CUP corpus callosum used for ROI analysis of tdT+ oligodendroglia. Scale bars, 50 μm. (**H**) Quantification of percentage of tdT+Olig2+ oligodendroglia in *B2m*$^{tdT}$ and *Cd74*$^{tdT}$ brain of CNS inflammatory models. Data points are an individual animal mean across all regions analyzed. n=3–9 mice/group. Unpaired Mann-Whitney t-test. Data represented are means ± s.e.m. (**I**) Regional ratio of tdT+Olig2+ oligodendroglia in four regions analyzed in *B2m*$^{tdT}$ brain. For each individual animal total tdT+Olig2+ cells in each region taken as a proportion of all tdT+Olig2+ oligodendroglia across all brain regions for that animal. n=3–9 mice/group. Data represented are means ± s.e.m. AT - adoptive transfer, AT-CUP - adoptive transfer cuprizone, CUP - cuprizone, ROI - region of interest.

The online version of this article includes the following video, source data, and figure supplement(s) for figure 8:

**Source data 1.** Images from confocal Z-stack from Imaris reconstruction of *Figure 8—video 1*.

**Source data 2.** Immunohistochemistry quantification, sheets are labeled with letter corresponding to data panels in *Figure 8*.

**Figure supplement 1.** Correlation of experimental autoimmune encephalitis (EAE) score with tdT+ oligodendroglia in EAE brain and regional ratio of tdT+ oligodendroglia across brain regions of central nervous system (CNS) inflammatory models.

**Figure supplement 1—source data 1.** Immunohistochemistry quantification, sheets are labeled with letter corresponding to data panels in *Figure 8—figure supplement 1*.

**Figure 8—video 1.** Imaris 3D reconstruction of *Cd74*$^{tdT}$ MOG$_{35-55}$ experimental autoimmune encephalitis (EAE) ventral brain confocal Z-stack.
https://elifesciences.org/articles/82938/figures#fig8video1

callosum in cuprizone and adoptive transfer cuprizone models (*Baxi et al., 2015*), and tdT+ oligodendroglia may be concentrated in areas of high inflammatory activity, we analyzed the regional means across the brain regions in these models.

The ventral brain demonstrated an enrichment of tdT+Olig2+ oligodendroglia in AT-CUP compared to cuprizone in *B2m*$^{tdT}$ mice (*Figure 8—figure supplement 1B*). *Cd74*$^{tdT}$ mice had a significant increase in corpus and corpus horn tdT+Olig2+ oligodendroglia in AT-CUP compared to cuprizone (*Figure 8—figure supplement 1C*). To control for the variability of tdT+ oligodendroglia in different *B2m*$^{tdT}$ animals, due to various clinical scores and degree of inflammatory activity, we calculated the regional ratio of tdT+ oligodendroglia as a percentage of total tdT+ oligodendroglia in an individual animal (*Figure 8I*). *B2m*$^{tdT}$ mice demonstrated a significantly higher proportion of tdT+ oligodendroglia in the corpus horn of cuprizone mice compared to EAE and AT-CUP mice (*Figure 8I*, *Figure 8—figure supplement 1D*), suggesting that tdT+ oligodendroglia are enriched in areas of high inflammatory activity.

## Isolation of MHC reporter expressing cells reveals distinct transcriptional subpopulations

To further define the properties of MHC expressing 'immune' oligodendroglia, we performed single-cell RNA sequencing (scRNA-seq) of tdT-positive and -negative cells sorted from *Cd74*$^{tdT}$ and *B2m*$^{tdT}$ MOG$_{35-55}$ EAE brains (*Figure 9*). Single cells were sorted from whole brains based on tdT expression and viability (*Figure 9—figure supplement 1A and B*) and 10× scRNA-seq was performed on three isolated populations (tdT positive and negative from *Cd74*$^{tdT}$ mice, and tdT positive from *B2m*$^{tdT}$ mice) with an average sequencing depth of 2231–2835 genes per cell. Clustering was validated by assessing known unique transcripts associated with each cell population (*Figure 9—figure supplement 1C*). Clustering tdT-positive and -negative populations from *Cd74*$^{tdT}$ mice revealed mostly non-overlapping clusters (*Figure 9A*). tdT-negative cells were comprised predominantly of T cells, endothelial cells, oligodendroglia, and granulocytes, and tdT-positive cells were predominantly microglia, monocytes/macrophages, dendritic cells, and B cells (*Figure 9B*). The tdT-positive sorted sample contained high *Cd74* and *tdT* transcript levels (*Figure 9C*). tdT-positive sorted cells from *B2m*$^{tdT}$ mice were predominantly monocytes/macrophages, microglia, endothelial cells, and T cells (*Figure 9D and E*), which varied in their levels of *tdT* and *B2m* transcripts (*Figure 9F*). There were very few astrocytes and neurons sequenced, likely due to difficulty in isolating these populations with papain dissociation (*Lo et al., 2021*). The *Cd74*$^{tdT}$ monocyte/macrophage cluster had distinct separation of tdT-positive and tdT-negative sorted cluster populations (*Figure 9A*) and differential gene analysis revealed enrichment of MHC class II, M2 activation, and macrophage infiltration chemokine, inflammatory signaling receptor, complement and lysosomal protease transcripts in tdT-positive myeloid cells (*Figure 9—figure supplement 2*, *Figure 9—figure supplement 2—source data 1*). These data indicate that

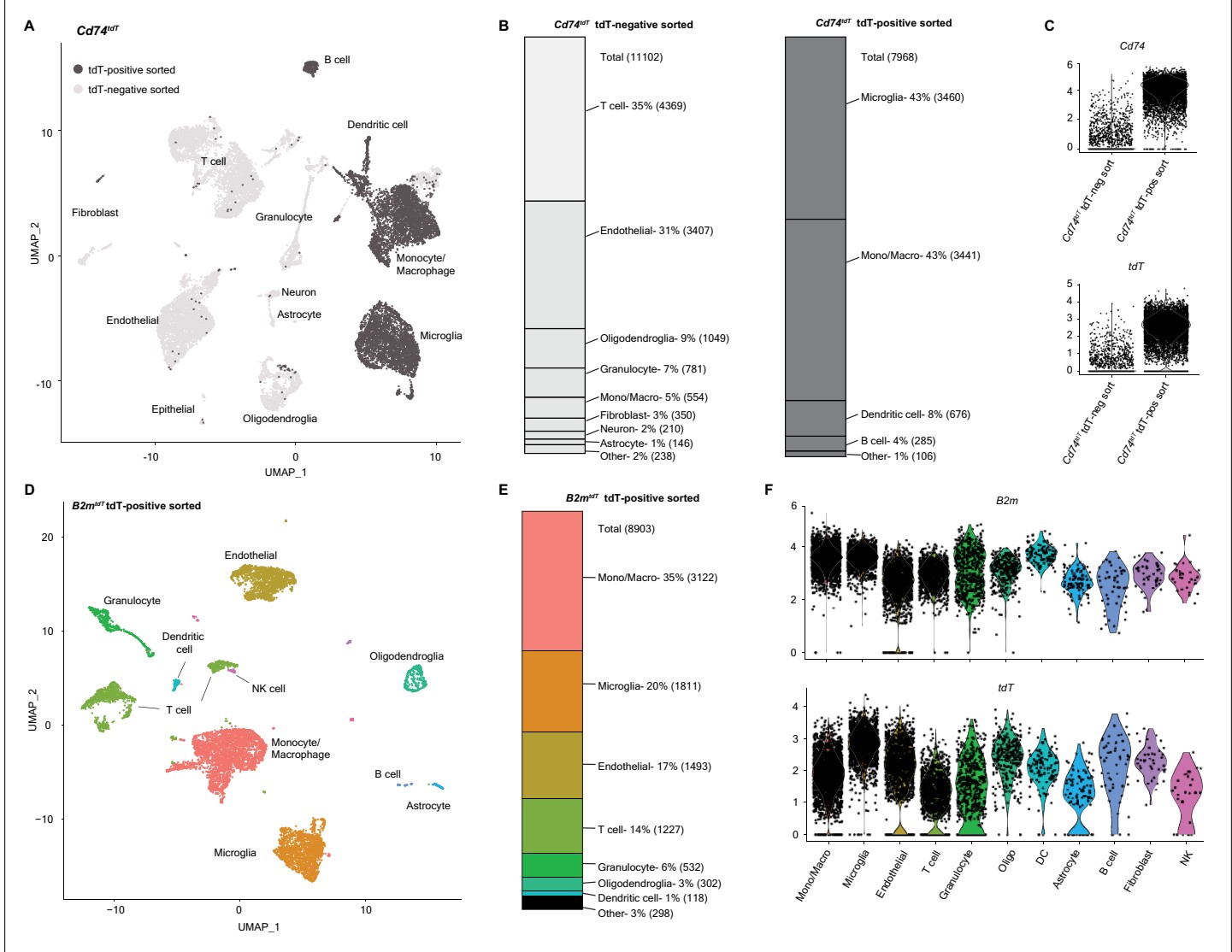

**Figure 9.** Single-cell RNA sequencing (scRNA-seq) of *Cd74^tdT* and *B2m^tdT* MOG_{35-55} experimental autoimmune encephalitis (EAE) brain. (**A**) Uniform Manifold Approximation and Projection (UMAP) *Cd74^tdT* MOG_{35-55} EAE brain scRNA-seq tdT-positive sorted cells (dark gray) and tdT-negative sorted cells (light gray). n=3 pooled *Cd74^tdT* EAE animals. (**B**) Cell identity percentages in *Cd74^tdT* tdT-positive and tdT-negative sorted cells, total cell count in parentheses. (**C**) Violin plots of log normalized expression levels of *Cd74* and *tdT* transcripts in tdT-positive and tdT-negative sorted samples. (**D**) UMAP *B2m^tdT* MOG_{35-55} EAE brain scRNA-seq TdT-positive sorted cells. n=3 pooled *B2m^tdT* EAE animals. (**E**) Cell identity percentages in *B2m^tdT* tdT-positive sorted cells, cell count in parentheses. (**F**) Violin plots of log normalized expression levels of *B2m* and *tdT* transcripts in *B2m^tdT* tdT-positive sorted cell clusters. DC - dendritic cells; Mono/Macro - monocyte/macrophage; NK - natural killer.

The online version of this article includes the following source data and figure supplement(s) for figure 9:

**Source data 1.** QC metrics for single-cell RNA sequencing data.

**Figure supplement 1.** Single-cell RNA sequencing (scRNA-seq) gating strategy and validation of clusters.

**Figure supplement 2.** *Cd74^tdT* MOG_{35-55} experimental autoimmune encephalitis (EAE) brain single-cell RNA sequencing (scRNA-seq) monocyte/macrophage cluster differential transcript expression levels.

**Figure supplement 2—source data 1.** Differential gene analysis data from myeloid/macrophage tdT-positive and tdT-negative clusters.

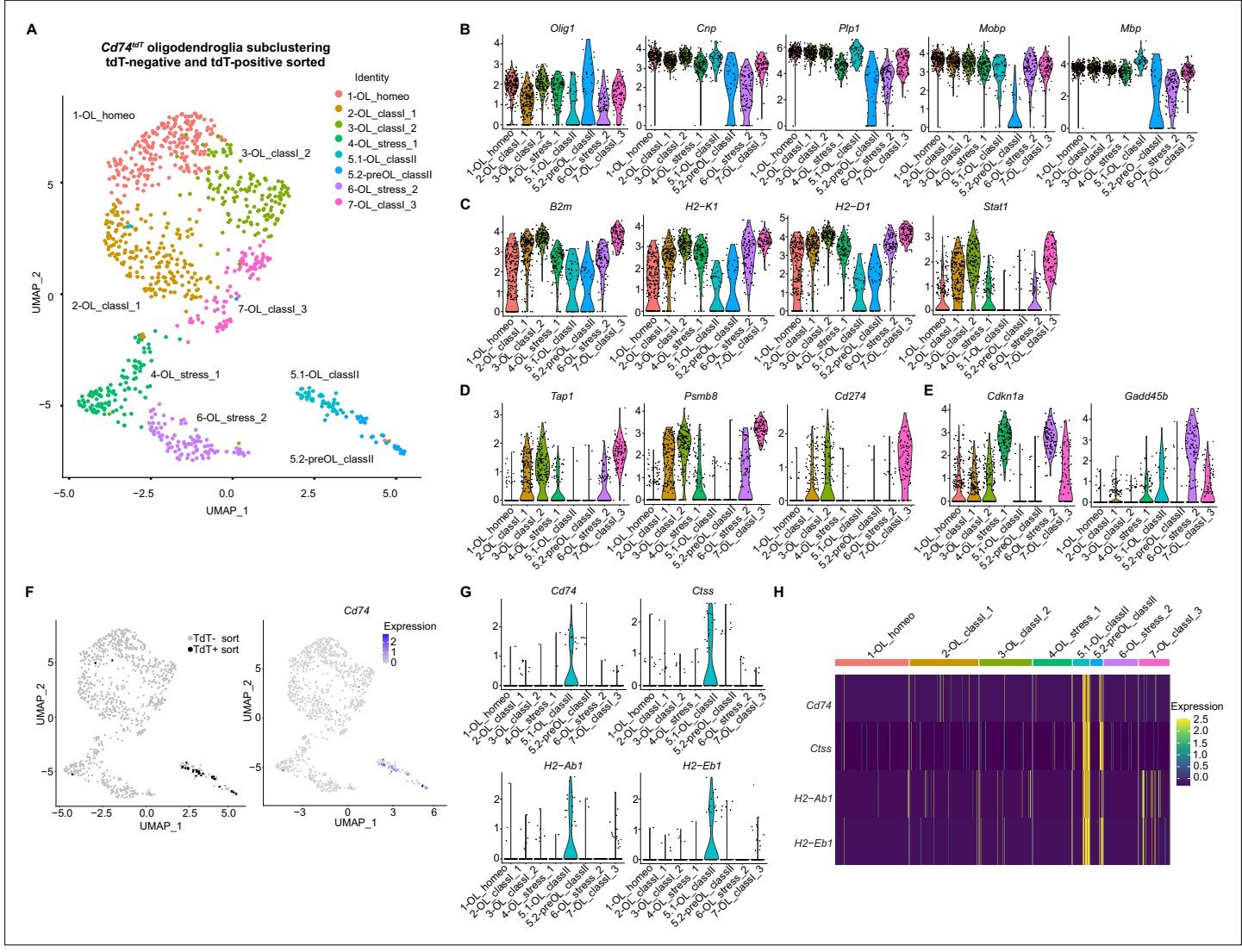

**Figure 10.** Oligodendroglial subclustering of single-cell RNA sequencing of $Cd74^{tdT}$ MOG$_{35-55}$ experimental autoimmune encephalitis (EAE) brain. (**A**) Uniform Manifold Approximation and Projection (UMAP) $Cd74^{tdT}$ MOG$_{35-55}$ EAE brain tdT-positive and tdT-negative sorted cells with eight oligodendroglial subclusters. (**B**) Violin plots of oligodendroglial markers *Olig1*, *Cnp*, *Plp*, *Mobp*, and *Mbp* transcript levels in oligodendroglial subclusters. (**C**) Violin plots of major histocompatibility complex (MHC) class I (*B2m*, *H2-K1*, *H2-D1*), and Stat1 transcript levels. (**D**) Violin plots of antigen transport (Tap1), immunoproteasome (*Psmb8*), and *Cd274/PD-L1* transcript levels. (**E**) Violin plots of cell cycle arrest *Cdkn1a* and *Gadd45b*. (**F**) UMAP indicating tdT-positive and tdT-negative sorted cells and *Cd74* transcript expression levels. (**G**) Violin plots of MHC class II transcripts (*Cd74*, *Ctss*, *H2-Ab1*, *H2-Eb1*). (**H**) Heatmap of MHC class II (*Cd74*, *H2-Ab1*, *H2-Aa*, *H2-Eb1*) expression levels in oligodendroglial subclusters.

The online version of this article includes the following source data and figure supplement(s) for figure 10:

**Source data 1.** Differential gene analysis data of oligodendroglial subclusters depicted in *Figure 10*.

**Figure supplement 1.** Oligodendroglial subclustering with contaminating clusters.

**Figure supplement 1—source data 1.** Differential gene analysis of each contaminating cluster compared to other oligodendroglial subclusters depicted in *Figure 10—figure supplement 1*.

MHC upregulation occurs in a minority of oligodendroglia, or that oligodendroglia that upregulate MHC are destroyed.

## Oligodendroglia subpopulations in EAE

Subclustering oligodendroglial cells from the $Cd74^{tdT}$ EAE brain (*Figure 10A*) resulted in nine subclusters (*Figure 10—figure supplement 1A*). On further analysis, several of the clusters with oligodendroglial transcripts were identified as T cells, myeloid cells, and endothelial cells (*Figure 10—figure*

*supplement 1B–C*). The presence of myelin transcripts in these cells may arise through phagocytosis of myelin debris, which has been reported during development (*Irfan et al., 2022*; *Nemes-Baran et al., 2020*). After removal of these contaminating clusters, 1082 oligodendroglia were distributed between eight subclusters (*Figure 10A*). Differential gene analysis revealed that pan-oligodendroglial marker *Olig1* and mature myelin transcripts (*Cnp, Plp, Mbp,* and *Mobp*) were present in all clusters, with the lowest levels of mature myelin markers and highest level of Olig1 in cluster 5.2 (*Figure 10B*) which was defined 'pre-OL' for intermediate oligodendrocyte, and other clusters labeled 'OL' as mature oligodendrocytes. Three clusters (2,3,7) labeled 'class 1' were notable for the highest levels of MHC class I receptor transcripts, as well as interferon-responsive *Stat1* transcription factor (*Figure 10C*), immunoproteasome *Psmb8*, peptide transport *Tap1* and *Cd274* transcripts (*Figure 10D*). It is possible that expression of *Cd274/PD-L1*, a MHC class I inhibitory molecule, prevents CD8-mediated killing of these oligodendroglia. Cell cycle arrest transcript *Cdkn1a* was highly expressed in two clusters (4,6) labeled 'stress' (*Figure 10E*), and growth arrest/apoptosis transcript *Gadd45b* was relatively enriched in one of these clusters (6). *Cd74^{tdT}* tdT-positive sorted oligodendroglia represented 3% (34/1082) of total oligodendroglia and were found predominantly in cluster 5.1 and 5.2, which correlated with *Cd74* transcript expression (*Figure 10F*). Cluster 5.1 and 5.2 were labeled 'class II' and MHC class II transcripts (*Cd74, Ctss, H2-Ab1, H2-Eb1*) were enriched in these clusters (*Figure 10G–H*).

To compare our iOPC/iOL EAE subsets with a previously published mouse EAE spinal cord dataset from *Falcão et al., 2018*, we integrated both datasets and repeated oligodendroglial subclustering (*Figure 11*). Oligodendroglial subclusters from both datasets overlapped to a high degree (*Figure 11A*), but with several exceptions. Our dataset consisted predominantly of mature oligodendrocytes and did not have *PDGFRa* transcript containing cells present in OPC clusters of Falcao dataset (*Figure 11—figure supplement 1A*). This was also reflected in the lack of cells in our dataset in the Uniform Manifold Approximation and Projection (UMAP) region corresponding to Falcao et al. OPC clusters (*Figure 11B*). The Falcao et al. dataset utilized an OPC reporter and lineage tracing approach, which may have resulted in an enrichment of OPCs, as opposed to our dataset which did not sort on reporter positive oligodendroglia. One mature Falcao et al. subset, MOL3_EAE, was under-represented in our dataset, which may reflect differences in spinal cord oligodendroglia compared to brain oligodendroglia. One of our subclusters, 1-OL_homeo, corresponded to primarily oligodendroglia enriched in naïve non-immunized mice from Falcao et al. dataset, which may represent homeostatic oligodendrocytes with low levels of MHC class I, MHC class II, and cell stress/apoptosis transcripts (*Figure 10C–H*). We selected upregulated transcripts from each oligodendroglial subcluster of our dataset (*Figure 10—source data 1*) and evaluated expression of these transcripts in corresponding subclusters in Falcao et al. dataset (*Figure 11C*). Transcripts highly expressed in our oligodendroglial subclusters demonstrated high expression in Falcao et al. subclusters, based on UMAP location. Together, this analysis provides further evidence of consistent MHC upregulation by subsets of oligodendroglia in the context of inflammation.

## Discussion

MHC upregulation by oligodendroglia in response to inflammation may influence cell survival, myelin repair, and disease progression, but the properties of these cells, their incidence, and ultimately their fate have been difficult to study in the CNS. To overcome this limitation, we developed two new mouse lines in which the fluorescent protein tdT is expressed under control of either the *B2m* or *Cd74* promoters, allowing identification of cells that upregulate MHC I and MHC II pathways, respectively. Our studies reveal that MHC class I and II reporter mice enable reliable detection of immune oligodendroglia in the CNS. Quantitative analysis from naïve and inflammatory disease model mice reveals that these glial cells represent a small, but diverse population that increase in the inflammatory environment, mirroring single-nucleus RNA sequencing results from human MS post-mortem tissues (*Schirmer et al., 2019*; *Jäkel et al., 2019*; *Absinta et al., 2021*). Although not restricted to lesion sites, immune oligodendroglia were more prevalent in areas of enhanced inflammation specific to disease pathology, where they have the potential to influence the behavior of both peripheral and central immune cells and subsequent remyelination. Future, longitudinal intravital imaging of *Cd74^{tdT}* and *B2m^{tdT}* mice will help to define the onset and persistence of MHC activation in these cells and determine if they are at higher risk for removal from the CNS through cytotoxic CD8 T cell-induced death.

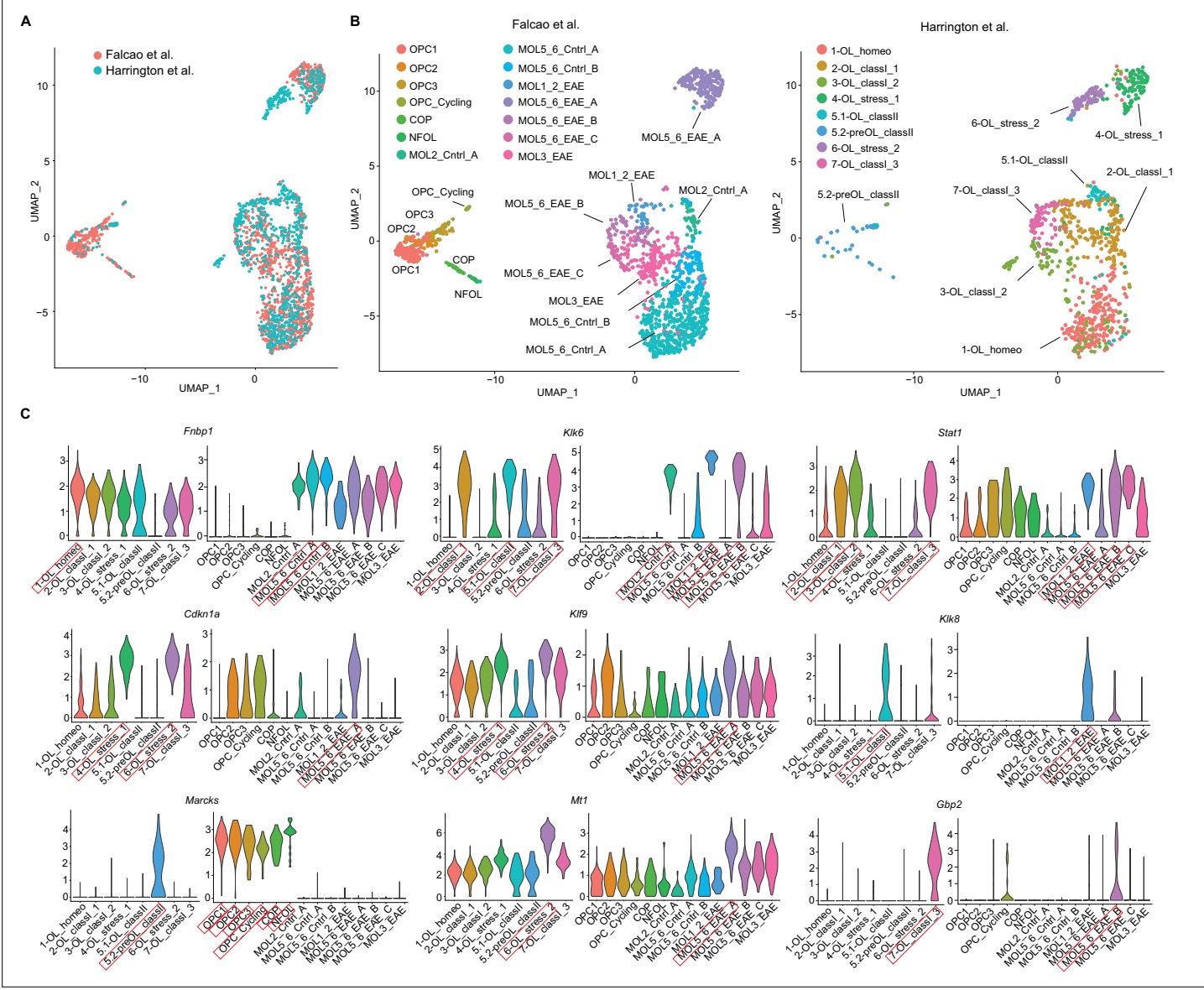

**Figure 11.** Integration of single-cell RNA sequencing experimental autoimmune encephalitis (EAE) oligodendroglial cluster dataset with previously published Falcao et al. EAE oligodendroglial cluster dataset. (**A**) Uniform Manifold Approximation and Projection (UMAP) of subclustering of combined datasets of *Cd74^tdT* oligodendroglial subclusters from EAE brain and Falcao et al. OPC reporter line oligodendroglial subclusters from EAE spinal cord. (**B**) UMAP of oligodendroglial subclustering with Falcao et al. dataset on left with labels of subclusters as previously published, and our dataset on right with corresponding *Figure 10* labels. (**C**) Violin plots of differentially expressed transcripts from each of our subclusters and corresponding transcript expression in Falcao et al. dataset. Red boxes indicate subcluster with highest expression and corresponding subcluster in Falcao et al. dataset based on UMAP location in panel B.

The online version of this article includes the following figure supplement(s) for figure 11:

**Figure supplement 1.** Differential expression of oligodendroglial transcripts in our experimental autoimmune encephalitis (EAE) oligodendroglial subclusters and Falcao et al. oligodendroglial subclusters.

Recent evidence suggests that MHC expression may vary with age (*Nemes-Baran et al., 2020*; *de la Fuente et al., 2020*; *Mishra et al., 2020*), and our scRNA-seq data revealed that a subset of MHC class I expressing oligodendroglia express *PD-L1/Cd274* transcripts, which may protect these cells from CD8 T cell-mediated death and may allow MHC class I oligodendroglia to persist in inflammatory settings. Whether these oligodendroglia exhibit impaired differentiation and capacity for remyelination remains to be determined. Conversely, oligodendroglia expressing MHC class II may be more prone to apoptosis or senescence through Gadd45b signaling (*Zaidi and Liebermann, 2022*).

Moreover, the frequency of immune oligodendroglia was correlated with disease severity, suggesting that these cells have a relationship with behavioral presentation that could be used to track the progression and treatment of demyelinating disease. The ability to isolate immune oligodendroglia and other cells with activated MHC pathways from the CNS will help define the phenotypic changes they exhibit in diverse disease and injury contexts, and provide new insight into the complex cellular interactions that modify disease progression and repair.

The value of these MHC reporter mice extends beyond OL lineage cells. Reactive microglia and astroglia can also express MHC class II and have been implicated in many neurodegenerative diseases. *Cd74* is a signature of disease-associated microglia in Alzheimer's (*Bryan et al., 2008*; *Swanson et al., 2020*), as well as being found in microglia associated with MS in post-mortem tissues (*Absinta et al., 2021*). Delineation of the transcriptional profiles of these cells can now be achieved by sorting tdT+ microglia, and their regional dynamics defined through in vivo two-photon imaging. MHC I induction has also been reported in neurons, where it can influence maturation and synaptogenesis (*Neumann et al., 1995*; *Corriveau et al., 1998*; *Huh et al., 2000*; *Goddard et al., 2007*; *Shatz, 2009*; *Elmer and McAllister, 2012*) and is upregulated in pathological states, such as viral infection and traumatic injury (*Neumann et al., 2002*; *Meuth et al., 2009*; *Joseph et al., 2011*). These reporter mice will facilitate analysis of neuronal MHC signaling in other injury and disease contexts, expanding our knowledge about the prevalence and impact of MHC signaling in the CNS.

## Materials and methods

### Mice

*2D2* (C57BL/6-Tg(Tcra2D2,Tcrb2D2) 1Kuch/J Jax stock #006912), *OT-II* (B6.Cg-Tg(TcraTcrb)425Cbn/J Jax stock #004194) and wild-type C57BL/6 mice were purchased from Jackson Laboratories. Generation of *B2m^tdT* and *Cd74^tdT* reporter lines is described below. All experiments were performed when mice were 8–16 weeks of age and both male and female mice were used. All animal procedures were performed according to protocols approved by the Johns Hopkins Animal Care and Use Committee.

Generation of *B2m^tdT* and *Cd74^tdT* reporter lines by Crispr/Cas9 Benchling gRNA design tool was used to select guide RNA (gRNA) sequences to target replacement of the stop codon with reporter repair construct. gRNA sequences were synthesized by Integrated DNA Technologies. Repair reporter constructs were cloned by Genscript and consisted of pUC57 vector with 500 bp homology arms of endogenous *B2m* and *Cd74* locus flanking *P2A-TdTomato-WPRE-pA* sequence (repair construct,- *Figure 1—source data 1*). Johns Hopkins Transgenic Core Laboratory injected C57BL/6 embryos with gRNA and repair construct (6 kb plasmid with 500 bp homology arms and 2339 knock-in sequence). Founder pups (62 for *B2m^tdT* and 61 for *Cd74^tdT*) were screened with primers spanning the 5' homology arm and tdTomato (out-L-F/TdT-R), 3' homology arm and WPRE sequence (WPRE-F/Out-R-R) and full-length insert 5' homology arm and 3' homology arm (In-L-F/In-R-R). For *B2m^tdT*, five founder pups demonstrated amplification of all primer sets and two of these founder pups had strong TdTomato endogenous fluorescence on peripheral blood flow. Both founder animals were bred to C57BL/6 to generate founder lines and both founders had identical reporter expression in all tissues analyzed. Full-length sequencing of insert was unsuccessful due to redundancy in tdTomato sequence. The founder with the strongest PCR amplification band of full-length insert was used for experiments in this study. For *Cd74^tdT*, one founder pup demonstrated amplification of all primer sets and had strong tdTomato endogenous fluorescence on peripheral blood flow. This founder animal was bred to C57BL/6 to generate the *Cd74^tdT* founder line.

### Oligodendrocyte progenitor culture

Oligodendroglial cultures were performed in three independent experiments and for each experiment one litter of pups per reporter was screened for tdT reporter expression and tdT-positive pups were pooled for immunopanning isolation. Post-natal P6-10 *B2m^tdT* and *Cd74^tdT* pups were screened for tdTomato expression with peripheral blood flow. tdT-positive pups were cervically decapitated, forebrains were dissected in HBSS, and three to five forebrains were pooled and gently chopped several times with a razor. Dissociation was performed with Miltenyi Neural Tissue dissociation kit P (130-092-628) and according to the manufacturer's protocol. After dissociation, cells were resuspended in 0.02% BSA in HBSS. Three 15 cm non-coated Petri dishes for each dissociation of three

to five pups were pre-incubated overnight at 4°C the day prior to dissociation. Plate coating: goat anti-rat IgG (Jackson 112-005-167) 1:333, goat anti-mouse IgG (Jackson 115-005-003) 1:333, and BSL1 (Vector L1100) 1:1000 all in Tris-HCl pH 9.5. Prior to dissection and dissociation, secondary antibody plates were washed with HBSS and incubated with primary antibody mouse CD11b (Bio-Rad MCA275G) and rat PDGFRa (BD Pharmigen 558774) at room temperature for at least 2 hr. BSL1 plate was equilibrated at room temperature. After cell dissociation, panning plates were washed with HBSS and cell suspension was added to BSL1 plate (negative selection for endothelial cells) for 10 min. Plate was tapped to dislodge loosely bound cells and unbound suspension was transferred to CD11b plate (negative selection for microglia) for 20 min and then PDGFRa plate (positive selection for oligoden-drocyte progenitors) for 90 min. To harvest bound cells, PDGFRa plate was washed with HBSS and 0.0625% trypsin in HBSS was added for 10 min at 37°C. FBS was used to dislodge cells and inhibit trypsin and cells were resuspended in OPC proliferation media. Cells were plated at a density of 50 k/coverslip with pre-plating to ensure adherence by adding 50 k cells in 50 µl media to center of cover-slip and incubating 37°C for 10 min then gently adding 450 µl of media to side of well. Coverslips were pre-coated with 10 µg/ml PDL in $dH_2O$ for several hours and washed with $dH_2O$ prior to cell plating. OPC proliferation media: DMEM high glucose pyruvate (Thermo Fisher 11995073), 100 U/100 µg/ml penicillin/streptomycin, 5 µg/ml N-acetyl-L-cysteine, 1×SATO (1 µg/ml human apo-transferrin, 1 µg/ml BSA, 0.16 µg/ml putrescine, 0.6 mg/ml progesterone, 0.4 ng/ml sodium selenite), 1xB27 (Thermo Fisher 17504044), 1xTrace Elements B (Cellgro 99-175C), 5 µg/ml insulin, 10 ng/ml d-biotin, 4.2 µg/ml forskolin, 20 ng/ml PDGFAA (PeproTech 100-13A), 1 ng/ml NT-3 (PeproTech 450-03), 10 ng/ml CNTF (PeproTech 450-13). Cells were maintained in 37°C incubator with 10% $CO_2$ and 50% of media was changed every 48 hr. Twenty-four hr post-plating, 50 ng/ml IFN-γ was added to media and after 72 hr of IFN-γ treatment, coverslips were fixed with 4% PFA for 10 min then washed twice in PBS.

## Western blotting

Tissues from a total of five wild-type, four $B2m^{tdT}$ and $Cd74^{tdT}$ adult mice were used for western blot analysis. Mice were anesthetized with isoflurane and cervically dislocated. Brain, spinal cord, spleen, and lymph nodes were dissected and transferred to cold RIPA lysis buffer with complete protease inhibitor cocktail (MilliporeSigma 4693116001) and mortar and pestle was used to grind tissue. Homogenate was centrifuged for 20 min 4°C at 12,000 rpm, supernatant was collected and protein concentration was measured with BCA assay (Pierce BCA protein assay kit 23225). Twenty µg of protein samples in Laemmli buffer were reduced with 0.1 M DTT and incubated at 100°C for 5 min. Samples and Chameleon Duo ladder (Li-Cor 928-60000) were loaded in 4–20% Mini-Protean TGX gel (Bio-Rad) and run at 200 V for 45 min. Wet transfer was performed onto PVDF membrane at 100 V for 60 min. Membranes were blocked in Odyssey blocking buffer (Li-Cor 927-60001) for 1 hr at room temperature then incubated in primary antibody in Odyssey blocking buffer with 0.2%Tween-20 over-night at 4°C. Primary antibodies were all used at 1:10,000 concentration: rabbit B2m (Dako A0072), goat tdTomato (MyBiosource MBS448092), rabbit CD74 (Abcam ab245692), and mouse beta-actin (Sigma A2228). Membranes were washed 3× in TBST (TBS 0.1%Tween-20) for 10 min. Membranes were incubated with secondary antibodies in Odyssey blocking buffer with 0.2%Tween-20 for 1 hr at room temperature. Secondary antibodies were all used at 1:10,000 concentration: goat anti-mouse IgG IRDye 680RD (Li-Cor 926-68070), goat anti-rabbit IgG IRDye 800CW (Li-Cor 926-32211), and donkey anti-goat IgG (Li-Cor 926-68074). Membranes were washed 3× in TBST for 10 min and imaged on LiCor Odyssey CLX system. Images were exported and band intensity was analyzed in ImageStu-dioLite software.

## OT-II CD4 T cell with $Cd74^{tdT}$ splenocyte co-culture assay

Spleens from five wild-type and five $Cd74^{tdT}$ adult mice were mashed over a 100 µm filter to create a single-cell suspension and red blood cells were lysed using RBC lysis buffer (eBioscience). Ovalbumin-specific naïve CD4+ T cells were isolated from OT-II mice using the Mojosort naïve CD4 T cell isolation kit (BioLegend), according to the manufacturer's protocols. Following isolation T cells were labeled with Cell Proliferation Dye efluor450 (eBioscience). Wild-type and $Cd74^{tdT}$ splenocytes were cultured at a ratio of 4:1 with OT-II CD4 T cells in Iscove's Modified Dulbecco's Medium (Gibco) with 10% fetal bovine serum (Gemini Bio-Products), 10 mM Glutamax (Thermo Fisher), 1 mM sodium pyruvate (Milli-poreSigma), 55 mM 2-mercaptoethanol (Gibco), and 100 U/100 µg/ml penicillin/streptomycin (Quality

Biological) with 10 µg/ml of ovalbumin. After 72 hr cells were harvested for flow cytometry. Cells were spun at 1500 rpm for 5 min, washed in PBS, and incubated with Fc block (BioLegend 156604) and live/dead Aqua (Thermo Fisher L34966) for 15 min. Cells were washed with FACs buffer and incubated antibodies at concentration of 1:100 in FACs buffer for 30 min. Antibodies: CD4 APC (BD 553051), vb5 FITC (BioLegend 139514), CD44 PECy7 (eBioScience 25-0441-82). Cells were washed in FACS buffer and ran on Cytek Aurora 4 laser flow cytometer. Compensation was performed with single stained UltraComp eBeads (ThermoScientific 01-3333-42), ArC amine reactive beads for viability dyes (LifeTechnologies A10346) and unstained co-cultures for tdTomato compensation. FlowJo software was used for analysis and gating. T cells were gated on singlet, cell, viable, CD4+, vb5+ then further subgated on proliferating with two or more divisions and CD44+. Each well was an independent experimental replicate.

## MOG$_{35-55}$ peptide EAE

For EAE experiments mice were obtained from four separate immunization experiments. A total of 5 clinical score 0 mice per reporter were taken at day 8 post-immunization and 10 $B2m^{tdT}$ score >0 and 11 $Cd74^{tdT}$ score >0 were sacrificed at day 8–17 post-immunization across the four separate experiments. One-fifth immunization experiment in which animals demonstrated a delayed EAE source was not analyzed. For immunization MOG$_{35-55}$ peptide was emulsified in 1:1 volume of MOG$_{35-55}$ 2 mg/ml in PBS and complete Freund's adjuvant (8 mg/ml mycobacterium tuberculosis in incomplete Freund's adjuvant) with syringes attached to stopcock for 10 min. Seventy-five µl of emulsion was injected subcutaneously on the left and right lateral abdomen for a total of 150 µg MOG$_{35-55}$ peptide. Pertussis toxin 250 ng was injected intraperitoneally on the day of immunization and 2 (dpi). Clinical scores were monitored daily after 7 dpi. The following criteria for clinical scoring was used from Hooke laboratories: 0.5 mild tail weakness, 1 limp tail, 1.5 limp tail and wobbly walk (no overt leg weakness), 2 limp tail and hindlimb weakness (not dragging), 2.5 limp tail and dragging of one or both hindlimbs but some movement at hindlimb, 3 limp tail and complete hindlimb paralysis (spinning or severe ataxia), 3.5 limp tail and complete hindlimb paralysis and hindlimbs held to one side of body or hindquarters flat, 4 all of 3.5 scoring with partial to complete front leg paralysis and minimal movement around the cage, 5 dead. Mice were sacrificed at 8–17 dpi.

## Cuprizone Th17 adoptive transfer

For cuprizone adoptive transfer a total of three separate adoptive transfer experiments were performed. A total of three cuprizone alone for $B2m^{tdT}$ and $Cd74^{tdT}$, four adoptive transfer alone for $B2m^{tdT}$ and $Cd74^{tdT}$, and six adoptive transfer-cuprizone for $B2m^{tdT}$ and eight for $Cd74^{tdT}$ were sacrificed across the three separate experiments. Mice were fed 0.2% cuprizone (bis(cyclo-hexanone) oxaldihydrazone [Sigma-Aldrich]) mixed with powdered, irradiated 18% protein rodent diet (Teklad Global) for 3.5 weeks, and chow was replaced every 2–3 days. CD4+ T cells were isolated from the spleens and draining lymph nodes of $2D2$ mice using the CD4+ isolation kit (BioLegend) and co-cultured with irradiated wild-type splenocytes at a ratio of 1:5. To polarize cells to a Th17 profile, 2.5 µg/mL anti-CD3 (BioLegend), 20 µg/mL anti-IL-4 (BioLegend), 20 µg/mL anti-IFN-γ (BioLegend), 30 ng/mLIL-6 (PeproTech), and 3 ng/mL TGFb (Thermo Fisher) were added to the medium IMDM (Thermo Fisher), fetal bovine serum (Gemini Bio-Products), penicillin and streptomycin (Quality Biologicals), 2-mercaptoethanol (Gibco), Glutamax (Thermo Fisher), and sodium pyruvate (MilliporeSigma). After 72 hr, cells were transferred to a new plate with fresh medium and IL-23 (R&D Systems). After 48 hr rest, cells were restimulated on plates coated with anti-CD3 and anti-CD28 (BioLegend). After 48 hr of restimulation, cells were collected and resuspended in PBS. Mice fed 0.2% cuprizone for 3.5 weeks (AT-CUP) followed by 3 days of cessation of cuprizone and no cuprizone treatment (AT only) mice were injected intraperitoneally with 10×10$^6$ Th17 cells. Clinical score was monitored daily starting at 7 days post-adoptive transfer and mice were sacrificed 7–17 days after adoptive transfer. Cuprizone without adoptive transfer and adoptive transfer without cuprizone was performed in parallel. For cuprizone alone mice were sacrificed 7 days after cessation of cuprizone diet. For adoptive transfer cuprizone clinical scores ranged from 0.5 to 3.5 with some animals demonstrating more atypical clinical scores. For adoptive transfer alone clinical scores ranged from 2.5 to 4.

## Immunohistochemistry

For baseline immunohistochemistry brain and spinal cord from four adult mice per reporter were analyzed. For all stains a minimum of 12 slides across three separate staining experiments were analyzed. For tissue harvesting mice were anesthetized with isoflurane and intracardially perfused with PBS followed by 4% PFA. Brain and spinal cord were dissected and post-fixed in 4% PFA for 24 hr then transferred to 30% sucrose for 2–3 days. Tissue was embedded in OCT (Tissue-Tek) and flash frozen on dry ice bath. Tissue was cut on Leica CM1850 cryostat and spinal cord 20 µm sections were mounted onto slides and brain 30 µm sections were mounted onto slides or collected in cryoprotectant (30% sucrose, 30% ethylene glycol, 10 mg/ml polyvinylpyrrolidone in 0.1 M PB buffer pH 7.4–10.9 mg/ml $Na_2PO_4$, 3.2 mg/ml $NaH_2PO_4$ in $dH_2O$). Floating sections were stored at –20°C and mounted sections were stored at –80°C. Floating sections were mounted onto slides prior to staining and allowed to dry for 10 min. Coverslip staining was performed in 24-well plate. Slides were washed with PBST (PBS Triton 0.5%) for 5 min, outlined with PAP pen (Thermo Fisher 5027627) and blocked with 5% normal goat serum (Vector S-1000) or 5% normal donkey serum (Jackson 017-000-121) in PBST for 1 hr. Slides were incubated in primary antibody in 5% normal goat or normal donkey serum in PBST over-night at 4°C. Primary antibodies: rabbit Olig2 1:1000 (MilliporeSigma AB9610), mouse Olig2 1:1000 (MilliporeSigma MABN50), mouse CC1/APC 1:500 (MilliporeSigma OP80), rabbit PDGFRa 1:500 (Cell Signaling 3174), rabbit Iba1 1:1000 (Wako 019-19741) goat Iba1 1:500 (Novus NB100-1028), goat TdTomato 1:2000 (MyBiosource MBS448092), rat IA/IE 1:100 (BD Pharmingen 556999), rabbit CD74 1:100 (Abcam ab245692), rabbit CD31 1:100 (Abcam ab28364), hamster CD11c 1:100 (Thermo Fisher 14-0114-82), TMEM119 (Abcam ab209064), rabbit NeuN 1:1000 (MilliporeSigma ABN78), mouse NeuN 1:500 (MilliporeSigma MAB377), mouse Parvalbumin 1:1000 (MilliporeSigma P3088), rabbit GFAP 1:1000 (Agilent Z0334), rabbit Sox9 1:1000 (Millipore AB5535), chicken GFAP 1:2000 (Aves), rat CD45 1:100 (Millipore 05-1416). Slides were washed 3× PBST then incubated with secondary antibody 1:500 in in 5% normal goat or normal donkey serum in PBST for 1 hr at room temperature. Secondary antibodies: goat anti-rabbit IgG 488 (ThermoScientific A11008) and 647 (ThermoScientific A21244), goat anti-rat IgG 488 (ThermoScientific A11006) and 647 (ThermoScientific A21247), goat anti-hamster IgG 488 (ThermoScientific A21110), goat anti-mouse IgG 488 (ThermoScientific A11001) and 647 (ThermoScientific A21235), donkey anti-goat IgG 544 (Abcam ab150134) and 647 (Thermo-Scientific A21447). Slides were washed 1× in PBST for 5 min, 2× PBS for 5 min, then coverslipped with Prolong Gold Antifade with DAPI (Life Technologies P36931).

## Cell quantification

For spinal cord analysis four lumbar sections with seven ROIs (central canal, bilateral dorsal horn, bilateral lateral white matter, and bilateral ventral white matter) were imaged per animal. For brain analysis four brain sections with seven ROIs (midline corpus callosum, bilaterally corpus horn, bilateral subventricular zone, and bilateral ventral brain) were imaged per animal. For ROIs epifluorescence Z-stack images were taken on Zeiss Axio Observer Z1 with 20× objective. The total number of Olig2+ cells and Olig2+tdT+ cells in each image were manually counted using the events feature in Zen Blue software. The mean percentage of tdT+Olig2+ cells was calculated by taking the proportion of total Olig2+tdT+ cells over the total Olig2+ cells across all ROIs in all sections. For spinal cord analysis the number of ROI images quantified ranged from 16 to 28 per animal and total number of oligodendrocytes counted ranged from 1341 to 7927 per animal. For brain analysis the number of ROI images quantified ranged from 16 to 28 per animal and total number of Olig2+ cells counted ranged from 1341 to 7927 per animal. For brain analysis the number of ROI images quantified ranged from 16 to 28 per animal and total number of Olig2+ cells counted ranged from 524 to 2860 per animal. To compare spinal cord lesion to non-lesion images, the presence of DAPI hypercellularity was used to categorize ROIs as lesion or non-lesion from eight to nine animals with clinical score >2 per reporter. The mean percentage of Olig2+tdT+ cells was calculated across lesion ROIs and non-lesion ROIs for these individual animals. The presence of DAPI hypercellularity was associated with CD45+ clusters when staining for CD45 was performed on additional sections for CD45 analysis. To evaluate inflammation on whole spinal cord sections compared to Olig2+tdT+ cells on that section tiled images of three to four lumbar spinal cord sections were taken from three animals per reporter with clinical score >2. The total number of Olig2+ and Olig2+tdT+ cells and percentage of area/density of CD45 staining were quantified in one tiled section. As an additional marker of inflammatory activity on an individual animal

basis the mean tdT fluorescent intensity of ROIs was calculated on non-thresholded 8-bit images using ImageJ. The individual animal mean tdT fluorescent intensity was calculated by taking the average of the mean tdT fluorescent intensities across all ROIs quantified for an individual animal. For PDGFRa spinal cord quantification additional PDGFRa staining was performed on six to eight animals per reporter with clinical score >2 and four $B2m^{tdT}$ pre-clinical score 0 animals. Spinal cord ROIs were imaged as described above and PDGFRa+Olig2+tdT+ and PDGFR-Olig2+tdT+ cells were counted. The mean percentage of PDGFRa+Olig2+tdT+ cells was calculated by taking the proportion of total PDGFRa+Olig2+tdT+ cells over the total Olig2+tdT+ cells across all ROIs in all sections. The number of Olig2+tdT+ cells counted ranged from 34 to 675 double positive cells per $B2m^{tdT}$ animal and 9–23 double positive cells per $Cd74^{tdT}$ animal.

## Microscopy

Epifluorescence images were taken on Zeiss Axio Observer Z1 with 20× objective. Tiled images were collected with multi-focal Z support points and stitched in Zeiss Zen Blue software. Confocal Z-stack images were taken on Zeiss 880 confocal with 40× objective in Johns Hopkins NINDS Multiphoton Imaging Core NS050274. Maximum intensity projections of Z-stacks were created in Zeiss Zen Black software. Imaris software was used for 3D video rendering of Z-stacks.

## Flow cytometry

For flow cytometry experiments three separate experiments were performed for each reporter line. Three $B2m^{tdT}$ were immunized with $MOG_{35-55}$ peptide and sacrificed at 15 dpi with clinical scores ranging 2.5–3. Four $Cd74^{tdT}$ animals were immunized with $MOG_{35-55}$ peptide and sacrificed at 12 dpi with clinical scores ranging 1.5–4 for one experiment and separate experiment three $Cd74^{tdT}$ animals were immunized with $MOG_{35-55}$ peptide and sacrificed at 13 dpi with clinical scores ranging 3–3.5. Non-immunized adult $B2m^{tdT}$ (n=9) and $Cd74^{tdT}$ animals (n=6) were sacrificed for baseline tissue for these experiments in parallel with EAE animals and as one separate experiment of baseline alone. Animals were anesthetized with isoflurane and intracardially perfused with cold HBSS. Spleens were dissected and dissociated by mashing over 100 µm filter and red blood cells were lysed using RBC lysis buffer (BioLegend 420301). Brain and spinal cords were dissected, cut with razor sagittally for two dissociations per animal, lightly chopped several times with fine razor, and transferred to pre-equilibrated enzyme solution at 37°C. Two separate dissociations with papain/DNase and collage-nase/DNase were performed for each sagittal half of the brain and spinal cord. Concentrations of enzymes: papain 20 U/ml (Worthington LS003119), collagenase IV 1 mg/ml (Worthington LS004188) and DNase 100 U/ml (Worthington LS002007). Papain buffer: EBSS (Sigma E7510), 22.5 mM D-glucose, 0.5 mM EDTA, 2.2 g/L NaHCO$_3$, 5.5 mM L-cysteine pH 7.4. Collagenase buffer: EBSS, 22.5 mM D-glucose, 2.2 g/L NaHCO$_3$, 3 mM CaCl$_2$ pH 7.4. Tissue dissociations were incubated at 37°C for 10 min followed by mechanical dissociation 10× with 1 ml pipette tip (first time with tip cut to larger bore) and repeated for a total of three incubations/triturations. Cell dissociations were filtered over a 70 µm filter, washed with PBS and myelin was removed with debris removal solution (Miltenyi 130-109-398) according to the manufacturer's protocol. Lower cell suspension layer was washed with PBS, filtered over FACS tube with 35 µm filter, incubated for 15 min with Fc block (BioLegend 156604) and live/dead Aqua (Thermo Fisher L34966) for spleen samples, washed with FACs buffer and incubated with cell surface antibodies at concentration of 1:100 in FACs buffer for 30 min. Cell surface antibodies: CD45 APCFire 750 (BioLegend 103211), Ly6G bv650 (BioLegend 127641), CD19 PacBlue (BioLegend 115523), CD11b PerCPCy5.5 (BioLegend 101228), CD11c PECy7 (BioLegend 117318), Clec12a APC (BioLegend 143406), CD86 bv421 (BioLegend 105035), CD40 FITC (BioLegend 124608), CD206 bv711 (BioLegend 141727), IA/IE bv785 (BioLegend 107645), CD3 PerCPCy5.5 (BioLegend 317336), CD8 PECy7 (BioLegend 100722), CD4 APC (BD 553051). For B2m staining, rabbit B2m 1:10 (Agilent A0072) primary was added to surface stain, and three washes were performed prior to secondary antibody staining with anti-rabbit bv421 1:500 (BioLegend 406410) with three additional washes after secondary antibody. For CD74 staining after surface stain, cells were fixed with IC fix (eBioscience88-8824-00) for 30 min followed by incubation with CD74 FITC 1:100 (BD 561941) in permeabilization buffer (eBioscience88-8824-00) for 30 min according to the manufacturer's protocol. Cells were washed in FACS buffer and ran on Cytek Aurora 4 laser (VBYR) flow cytometer in Johns Hopkins Ross Flow Cytometry Core. Compensation was performed with single stained UltraComp

eBeads (ThermoScientific 01-3333-42), ArC amine reactive beads for viability dyes (LifeTechnologies A10346) and unstained wild-type dissociated spleen, brain and spinal cord for tdT compensation. FlowJo software was used for analysis and gating. Spleen was gated on singlet, cell, viable, CD45+, Ly6G- then further subgated on CD19+ (B cells), CD11b+ (monocytes), CD11b-CD11c+ (dendritic cells). For brain and spinal cord EAE cells were gated on singlets, cell, CD45+CD11b+ and then further subgated on Clec12a+ vs Clec12a-. Both Clec12a+ (infiltrating myeloid) and Clec12a- (microglia) were gated on co-stimulatory molecule CD40 and CD86 expression. Collagenase and papain dissociations were compared without any notable differences and dissociations with the highest yield of cells of interest were presented (collagenase for $Cd74^{tdT}$ infiltrating myeloid cells and microglia and papain for $B2m^{tdT}$ microglia). Tissue from each animal was stained and analyzed separately for each biological replicate. Multiple flow staining panels were performed on biological replicate and the most informative panels were used for data analysis.

## Single-cell RNA sequencing

For single-cell experiments two separate experiments were performed, one for each reporter line. Three $B2m^{tdT}$ MOG$_{35-55}$ peptide immunized 16-week-old male animals were sacrificed at 13 dpi with clinical scores of 2.5, 4, and 4. Four $Cd74^{tdT}$ MOG$_{35-55}$ peptide immunized 15-week-old female animals were sacrificed at 14 dpi with clinical scores of 2.5, 3, 3.5, 3.5. Animals were anesthetized with isoflurane and intracardially perfused with cold HBSS without cations with 5 µg/ml actinomycin (Sigma A1410) and 10 µM triptolide (Sigma T3652). Brains and spinal cords were dissected and placed in six-well plate with cold dissection buffer (HBSS no cations with 5 µg/ml actinomycin, 10 µM triptolide, and 27 µg/ml anisomycin [Sigma A9789]) protected from light. Brains and spinal cords were lightly chopped several times with fine razor and transferred to pre-equilibrated at 37°C dissociation buffer with enzymes (papain 20 U/ml and DNase 100 U/ml with 5 µg/ml actinomycin, 10 µM triptolide, and 27 µg/ml anisomycin in EBSS, 22.5 mM D-glucose, 0.5 mM EDTA, 2.2 g/l NaHCO$_3$, 5.5 mM L-cysteine pH 7.4). Tissue dissociations were incubated at 37°C for 10 min followed by mechanical dissociation 10× with 1 ml pipette tip (first time with tip cut to larger bore) and repeated for a total of three incubations/triturations. Cell dissociations were filtered over a 70 µm filter, washed with PBS and myelin was removed with two rounds of debris removal solution (Miltenyi 130-109-398) according to the manufacturer's protocol. Cell pellet was resuspended in PBS and filtered over FACS tube with 35 µm filter and incubated with Zombie live/dead Violet (BioLegend 423114) for 15 min in cold PBS at 4°C, washed in cold PBS and resuspended in 0.5% BSA 1 mM EDTA PBS at concentration of 1×10$^6$ cells/ml in cold PBS on ice for single-cell sorting. For $Cd74^{tdT}$ samples, two animals were pooled for two sorts. For $B2m^{tdT}$ samples, all three samples were combined then divided into two tubes and only one tube was sorted. Only brain samples were sorted due to the level of myelin debris in spinal cord samples. Brain samples were sorted on Aria Ilu 3 laser (VBR) in Johns Hopkins Ross Flow Cytometry core and gated on scatter and viability. tdT-positive and tdT-negative populations were collected in 0.1% BSA PBS on ice. Samples were sorted for a limit of 2 hr before submission to Johns Hopkins sequencing core. Cell viability and concentration was checked on a Countess II automated hemocytometer using Trypan Blue (Life Technologies) exclusion. Cell volumes calculated to capture 10,000 cells were loaded into a Chromium Next GEM Chip G and GEMs (Gel Bead-in-emulsion) were generated using a Chromium Controller using Single Cell 3' v3 chemistry (10× Genomics). Barcoded single-cell libraries were then generated according to the manufacturer's recommendations. Libraries were sequenced on a NovoSeq 6000 (Illumina).

## scRNA-seq analysis

scRNA-seq data have been deposited in the NCBI's Expression Omnibus and are accessible through GEO Series accession number GSE213739. Sequencing code is provided as Source Code File 1. Sequencing data was aligned to the mouse genome (*Howe et al., 2021*) modified to include the *p2A-tdTomato* sequence (named *tdT* in the reference) using cellranger (v6.1.0, 10× Genomics). Filtered features from cellranger were imported into R (v4.1.2) using Seurat (*Satija et al., 2015*) (v4.1.0). Additional per cell metrics were assessed including gene count, UMI count, and ratio of reads mapping to mitochondrial genes. Cells were excluded for gene and UMI counts using adaptive thresholds (±3 median absolute deviations) with scuttle (*McCarthy et al., 2017*) (v1.4.0) or for mitochondrial gene ratios greater than 10%. The $Cd74^{tdT}$ tdT-positive sorted dataset had 8307 cells with median

gene count per cell of 2839 (2122,3478), median UMI count per cell of 11,400 (6522,16856), and median mitochondrial ratio of 3.149% (2.356%,4.196%) before filtering and 7968 cells with median gene count per cell of 2895 (2210,3508), median UMI count per cell of 11848 (6990,17132) and median mitochondrial ratio of 3.105% (2.333%,4.061%) after filtering. The *Cd74*^tdT tdT-negative sorted dataset had 12,091 cells before filtering with median genes per cell of 2241 (1736,2878), median UMI count per cell of 7014 (3828,11048), and median mitochondrial ratio of 3.115% (2.343%,4.228%). After filtering, the *Cd74*^tdT tdT-negative sorted dataset had 11,197 cells with median genes per cell of 2307 (1843,2940), median UMI per cell of 7467 (4292,11505) and median mitochondrial ratio of 3.025% (2.299%,3.925%). The *B2m*^tdT tdT-positive sorted dataset before filtering had 10,259 cells with median gene count per cell of 2626 (1812,3465), median UMI count per cell of 8416 (4030,14703), and median mitochondrial ratio of 3.707% (2.795%,5.150%). After filtering, the *B2m*^tdT tdT-positive sorted dataset had 8903 cells with median genes per cell of 2826 (2107,3580), median RNA per cell of 9868 (5374,15701), and median mitochondrial ratio of 3.514% (2.688%,4.547%). The *Cd74*^tdT tdT-negative sorted and *Cd74*^tdT tdT-positive sorted datasets were then merged into a single object prior to data normalization. For clustering purposes, UMI counts were normalized with a regularized negative binomial regression via the SCTransform function in Seurat (*Hafemeister and Satija, 2019*) using the glmGamPoi method (v1.6.0) (*Ahlmann-Eltze and Huber, 2021*). This was followed by principal component analysis (PCA) dimensionality reduction followed by UMAP dimensionality reduction on the first 40 principal components. Clustering was performed by creating a shared nearest-neighbor graph using the first 40 principal components followed by Louvain clustering. Cluster annotation was done in an automated fashion with SingleR (v1.8.1) (*Aran et al., 2019*) against the MouseRNASeqData set of 358 mouse bulk RNA-seq samples from sorted populations (*Benayoun et al., 2019*) available in the celldex package (v1.4.0). Automated annotations were then checked manually by comparing against well-characterized reference genes. For purposes of expression visualization (violin plots, feature plots), UMI counts were log normalized with Seurat. For heatmaps, normalized UMI counts were scaled with Seurat's ScaleData function. Subclustering analysis was performed by subsetting the cells of interest from the larger dataset (oligodendrocytes from merged *Cd74*^tdT tdT-positive sorted and *Cd74*^tdT tdT-negative sorted dataset, or oligodendrocyte cluster 5 from the olidogendrocyte dataset) and repeating above analysis on just the subsetted cells.

## Data integration and comparison

Raw counts for the previously published dataset by *Falcão et al., 2018*, were retrieved from the Gene Expression Omnibus under accession GSE113973. Annotation for the same dataset was retrieved from the UCSC cell browser (*Speir et al., 2021*) under accession 'oligo-lineage-ms' including which cells and genes were retained in the final published dataset as well as which cluster, mouse model, and condition each cell belonged to. The raw count data was then subsetted to only include the genes and cells that were present in the UCSC cell browser. The counts were then log normalized with Seurat and integrated with the oligodendrocyte subset of the merged *Cd74*^tdT dataset. First, the top 2000 shared variable genes were identified and used to find anchors with canonical correlation analysis (*Stuart et al., 2019*). The two datasets were then integrated with the Seurat function IntegrateData based on these anchors. The newly formed integrated dataset was then scaled and we performed PCA dimensionality reduction followed by UMAP reduction using the first 17 principal components.

## Statistical analysis

GraphPad Prism software was used for statistical analysis. For flow cytometry data unpaired t-tests were used to compare groups. For EAE lesion analysis unpaired Mann-Whitney t-test was used to compare groups given the variation in clinical scores combined into one group. For comparison of same animal lesion to non-lesion a paired Wilcoxon t-test was used. Simple linear regression was used for comparison of Olig2+tdT+ cells to EAE clinical score, mean tdT gray value, and mean percentage of area/density of CD45 staining.

## Acknowledgements

The authors thank Linsa Orzolek and Tyler Creamer of the JHMI Single Cell & Transcriptomics Core for assistance generating and sequencing scRNAseq libraries. The authors also thank JHU Ross Flow Cytometry Core for assistance with FACs and for the use of four laser Cytek Aurora. The Core Aurora

is funded by NIH grant S10OD026859. These studies were supported by grants from the NIH (NIA AG072305 to DEB, R01NS041435 to PAC), National Multiple Sclerosis Society (FAN-1707-28857 to EPH), National Science Foundation Graduate Research Fellowship to RBC, and the Dr. Miriam and Sheldon G Adelson Medical Research Foundation to DEB.

## Additional information

### Funding

| Funder | Grant reference number | Author |
| --- | --- | --- |
| National Institutes of Health | NIA AG072305 | Dwight E Bergles |
| National Multiple Sclerosis Society | FAN-1707-28857 | Em P Harrington |
| Dr. Miriam and Sheldon G Adelson Medical Research Foundation | | Dwight E Bergles |
| National Science Foundation | | Riley B Catenacci |
| National Institutes of Health | R01 NS041435 | Peter A Calabresi |

The funders had no role in study design, data collection and interpretation, or the decision to submit the work for publication.

### Author contributions

Em P Harrington, Conceptualization, Resources, Data curation, Formal analysis, Supervision, Validation, Investigation, Visualization, Methodology, Writing – original draft, Project administration, Writing – review and editing; Riley B Catenacci, Resources, Data curation, Formal analysis, Supervision, Validation, Investigation, Visualization, Writing – review and editing; Matthew D Smith, Data curation, Software, Formal analysis, Validation, Investigation, Visualization, Writing – review and editing; Dongeun Heo, Conceptualization, Resources, Validation, Methodology; Cecilia E Miller, Formal analysis, Investigation; Keya R Meyers, Investigation, Writing – review and editing; Jenna Glatzer, Visualization; Dwight E Bergles, Peter A Calabresi, Conceptualization, Resources, Supervision, Funding acquisition, Methodology, Project administration, Writing – review and editing

### Author ORCIDs

Em P Harrington ⓘ http://orcid.org/0000-0001-6352-8687
Cecilia E Miller ⓘ http://orcid.org/0009-0008-4455-436X
Jenna Glatzer ⓘ http://orcid.org/0000-0002-6809-9401
Dwight E Bergles ⓘ http://orcid.org/0000-0002-7133-7378

### Ethics

All animal procedures were performed according to protocols approved by the Johns Hopkins Animal Care and Use Committee protocol #MO22M158.

### Decision letter and Author response

Decision letter https://doi.org/10.7554/eLife.82938.sa1
Author response https://doi.org/10.7554/eLife.82938.sa2

## Additional files

### Supplementary files

• Transparent reporting form
• Source code 1. Code used to analyze single cell datasets.

## Data availability

Sequencing data has been deposited in GEO under the accession code GSE213739. Code used to analyze sequencing data is provided as Source Code 1. All data generated or analyzed during this study are included in manuscript source data files.

The following dataset was generated:

| Author(s) | Year | Dataset title | Dataset URL | Database and Identifier |
|---|---|---|---|---|
| Harrington EP, Catenacci RB, Smith MD, Heo D, Miller C, Meyers KR, Glatzer J, Bergles DE, Calabresi PA | 2022 | MHC class I and MHC class II reporter mice enable analysis of immune oligodendroglia in mouse models of multiple sclerosis | https://www.ncbi.nlm.nih.gov/geo/query/acc.cgi?acc=GSE213739 | NCBI Gene Expression Omnibus, GSE213739 |

The following previously published dataset was used:

| Author(s) | Year | Dataset title | Dataset URL | Database and Identifier |
|---|---|---|---|---|
| Castelo-Branco G, Falcão AM, van Bruggen D, Marques S, Agirre E, Meijer M, Jäkel S, Cacais AO, Vanichkina D, Floriddia EM, ffrench-Constant C, Williams A | 2018 | Disease-specific oligodendrocyte lineage cells express immunoprotective and adaptive immunity genes in multiple sclerosis | https://www.ncbi.nlm.nih.gov/geo/query/acc.cgi?acc=GSE113973 | NCBI Gene Expression Omnibus, GSE113973 |

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
