## [Editor Report]

This study reports an important new resource, MHC class I and MHC class II reporter mice, which provide a means to monitor MHC activation in vivo. The authors use these mice to study inflammatory demyelination in two mouse models of multiple sclerosis. The study provides a compelling demonstration of the new reporter lines as valuable tools for the analysis of inflammation and neurodegeneration.

---

## [Decision Letter]

**Decision letter after peer review:**

Thank you for submitting your article "MHC class I and MHC class II reporter mice enable analysis of immune oligodendroglia in mouse models of multiple sclerosis" for consideration by *eLife*. Your article has been reviewed by 3 peer reviewers, and the evaluation has been overseen by a Reviewing Editor and Carla Rothlin as the Senior Editor. The following individuals involved in the review of your submission have agreed to reveal their identity: Goncalo Castelo-Branco (Reviewer #1); Tim Vartanian (Reviewer #3).

Essential revisions:

1. The authors should explain the rationale for selecting B2m and Cd74, instead of other genes involved in these pathways.

2. MHC I has been traditionally described as expressed in all nucleated cells (https://www.ncbi.nlm.nih.gov/pmc/articles/PMC1783040/), although recent single-cell RNA-Seq studies suggest that this might not be the case. The data presented by the authors suggest that indeed activation of MHC I expression might be restricted, and it would be good if the authors discuss this aspect in relation to the current literature.

3. In Figure 1c, the difference observed in CD74-TdT is of much lower magnitude than B2m-TdT, could the authors comment on the reason why they think this is the case?

4. Line 162 – "B2m transcripts are upregulated in Cux2 expressing cortical layer II/II pyramidal neurons in human MS, which are susceptible to neurodegeneration3. Expression of MHC class I by these neurons may allow them to present antigen and render them more prone to death in inflammatory conditions." This is most likely referring to Supplementary Figure 4j, but no Cux2 staining to identify this cortical layer is provided. Also, it should be cortical layer II/III, not cortical layer II/II.

5. The authors perform several quantifications of the lesion and non-lesion areas, but only mention DAPI hypercellularity to define a lesion, more details should be provided. Also, how many lesions were quantified per mouse, and which regions of the brain and spinal cord? Also, whenever the authors present a high-magnification view of lesions (e.g. Figure 2), it would be good to present a lower magnification image of the same region, to orient the reader.

6. While the authors present a supplementary video to confirm the colocalization of TdT and oligodendrocyte markers, it would be good to present Z-stacks of the images in Figure 2c. Also, the authors should describe in the methods the parameters by which they define which cells are double positive.

7. The authors find that the % of TdT+ Olig2+ cells out of Olig2+ cells was around 0.6% in the spinal cord, while Falcao and colleagues (Nature Medicine 2018) found this percentage to be 3-4%. The authors can discuss possible reasons for these differences.

8. In line 222 and Figure 2F, the authors indicate that the number of OPCs expressing TdT is much lower than CC1+ positive cells. It would be interesting to discuss why this might be the case and the possible implications of this difference.

9. Figure 2I-J – The differences observed are borderline significant and very variable between lesions and non-lesions. Also in J, representative images are shown from different EAE scores.

10. Given that most of the EAE pathology occurs in the spinal cord, it is surprising that the authors present their single-cell studies with brain samples. In the Material and Methods, the spinal cord is also mentioned, with challenges in the sample preparation. The authors can integrate their dataset with other similar EAE datasets (Falcao et al., Nature Medicine 2018, Wheeler et al., Nature 2020) to compare the properties of the MHC1+ and/ or MHC2+ oligodendroglia (and astrocytes) they identify with the immune oligodendroglia and altered astrocytes in these datasets. Also, while the purpose of the single-cell experiment is to confirm the transcriptional identity of the TdT cells, which the authors succeed to do, it appears that n=1, and additional biological replicates would be recommended.

11. From Figure 3G and 3K, it appears that cluster 5 can be subclustered in 2 subpopulations, have the authors attempted to perform further subclustering?

12. The authors mention in the methods the redundancy in the TdT sequencing. Did this constitute a problem for the detection of the TdT transcripts in the single-cell analysis?

13. No code for the single-cell RNA-Seq analysis is provided and more information regarding QC (number of UMIs, % mitochondrial genes, etc) should be included.

14. Many of the images have low resolution and are thus difficult to examine. In particular, Figure 1 would benefit of higher resolution images for instance at the lesion areas, so the reader can observe the labelling in individual cells.

15. Figure 3—figure supplement 3 is completely missing.

16. Figure 3—figure supplement 4 – can the authors show the expression of additional oligodendrocytes markers, such as PLP1, CNP, Mog, Mag, and transcription factors as Olig1 and Olig2, to further assess which type of transcripts are contaminating/being phagocytized by non-oligodendroglial cells.

17. In Figure and Supplementary Figure legends, please state the n for each experiment shown and how many times the experiment was replicated.

18. The Material and Methods part regarding animals should include much more detail and follow the ARRIVE guidelines as much as possible. Please clarify the methods for the flow experiments described in 1 C-E. Was there an analysis of myelin?

19. In general, the manuscript is difficult to navigate, with much of the information in supplementary figures. We would recommend the authors reorganize the manuscript text and figures accordingly.

20. The presence of the 2A self-cleaving peptide should be highlighted when describing the reporter genes so that the reader can better understand reporter expression from the constructs.

21. The authors demonstrate the presence of the reporter fusion proteins on Western blot. Does the tomato portion need to be released from the fusion to provide fluorescence or is the presence of the fusion sufficient?

22. B2m and CD74 are necessary for cell surface expression of class I and II, respectively. Do the fusions influence (disrupt) cell surface class I or class II expression in reporter mice?

23. Some comment on astrocyte expression, or lack thereof, of the reporters in the inflammatory models examined would be of interest.

24. The mouse models report on class I and class II expression, which in turn report on the presence of inflammatory cytokines, particularly IFN-γ. Please comment on the spatial distribution of reporter-expressing cells in the CNS of their inflammatory models relative to the location of the immune cell infiltrates. This would provide some insight into the diffusion of the inflammatory cytokines away from the infiltrates.

25. Figure 1, please show boxed regions of interest for the merged images shown below F and G at high power so we can decern at a minimum leptomeninges from parenchyma and overlapping from non-overlapping fluorescence (probably ventrolateral cord).

26. The ventral brain/meningeal cluster would be better emphasized as a separate figure devoted to the brain.

27. It would be nice to show it in conjunction with the expression of CD74-TdT and B2m-TdT. Is the meningeal CD74-TdT expression in Figure 1 Supp 3F (baseline) more extensive than in Figure 1F (EAE score 0)?

28. In Figure 1 supp 4, B2m-TdT expression seems extensive in the cortex (more than just layer 2) and in the corpus callosum.

29. Are B2m-TdT and NeuN non-overlapping in the cortex (1 supp 4J)? If so, what lineage is B2m-TdT positive in 4J? What cell types are accounting for the intense B2m-TdT signal in the corpus callosum at baseline? What is the nature of the spindle-shaped cells that are Iba1 negative, B2m-TdT positive in the spinal cord (Figure 1 supp 4E)? What is the cell type in the CD74-TdT meningeal clusters in the brain (1I)?

30. In Figure 2, please provide the HPF of the B2m-TdT glial culture with IFNγ (2A). The TdT signal in 2B seems highly polarized relative to olig2 – is this a consistent finding? For 2C, it would be helpful to see the olig2 (green) unmerged images. Does the quantitation of TdT+/Olig2+ cells in the EAE spinal cord (2D) refer to total spinal cord or lesions? Similarly, for 2H, is this quantification lesion-based or total cord? How were ROI's determined for the quantitative data in Figure 2 (Please include this in the methods)? For 2F please make it somehow clearer that the y-axis is PDGFR (-).

31. Why was the brain used for the scRNA seq in Figure 3 as opposed to the spinal cord? Was this analysis done on samples at a particular EAE stage? In the results text can you either cite or explain bias based on the efficiency of generating viable single cells for different cell populations? Similarly, would the immune-oligodendroglial cluster potentially be less viable since it is already considered stressed and thus the relative size underestimated? How do the oligodendrocyte subclusters differ by CD74-TdT expression?

32. The discussion on the highest MHC I expressing oligodendrocyte subcluster is very interesting (proteosome, PD-L1) and I would recommend expanding this discussion since this cluster is the focus of the paper.

33. In the heat map (3K) I'm trying to understand what the individual vertical bars for a particular cluster and class II gene represent? Technical replicates? Experimental replicates? Perhaps p values are needed?

*Reviewer #1 (Recommendations for the authors):*

The manuscript is acceptable for publication in *eLife*, although there are several points the authors will need to address:

– The authors should explain the rationale for selecting B2m and Cd74, instead of other genes involved in these pathways.

– MHC I has been traditionally described to be expressed in all nucleated cells (https://www.ncbi.nlm.nih.gov/pmc/articles/PMC1783040/), although recent single-cell RNA-Seq studies suggest that this might not be the case. The data presented by the authors suggest that indeed activation of MHC I expression might be restricted, and it would be good if the authors discuss this aspect in relation to the current literature.

– In Figure 1c, the difference observed in CD74-TdT is of much lower magnitude than B2m-TdT, could the authors comment on the reason why they think this is the case?

– Line 162 – "B2m transcripts are upregulated in Cux2 expressing cortical layer II/II pyramidal neurons in human MS, which are susceptible to neurodegeneration3. Expression of MHC class I by these neurons may allow them to present antigen and render them more prone to death in inflammatory conditions." This is most likely referring to Supplementary Figure 4j, but no Cux2 staining to identify this cortical layer is provided. Also, it should be cortical layer II/III, not cortical layer II/II.

– The authors perform several quantifications of the lesion and non-lesion areas, but only mention DAPI hypercellularity to define a lesion, more details should be provided. Also, how many lesions were quantified per mouse, and which regions of the brain and spinal cord? Also, whenever the authors present a high-magnification view of lesions (e.g. Figure 2), it would be good to present a lower magnification image of the same region, to orient the reader.

– While the authors present a supplementary video to confirm the colocalization of TdT and oligodendrocyte markers, it would be good to present Z-stacks of the images in Figure 2c. Also, the authors should describe in the methods the parameters by which they define which cells are double positive.

– The authors find that the % of TdT+ Olig2+ cells out of Olig2+ cells was around 0.6% in the spinal cord, while Falcao and colleagues (Nature Medicine 2018) found this percentage to be 3-4%. The authors can discuss possible reasons for these differences.

– In line 222 and Figure 2F, the authors indicate that the number of OPCs expressing TdT is much lower than CC1+ positive cells. It would be interesting to discuss why this might be the case and the possible implications of this difference.

– Figure 2I-J – The differences observed are borderline significant and very variable between lesions and non-lesions. Also in J, representative images are shown from different EAE scores.

– Given that most of the EAE pathology occurs in the spinal cord, it is surprising that the authors present their single-cell studies with brain samples. In the Material and Methods, the spinal cord is also mentioned, with challenges in the sample preparation. The authors can integrate their dataset with other similar EAE datasets (Falcao et al., Nature Medicine 2018, Wheeler et al., Nature 2020) to compare the properties of the MHC1+ and/ or MHC2+ oligodendroglia (and astrocytes) they identify with the immune oligodendroglia and altered astrocytes in these datasets. Also, while the purpose of the single-cell experiment is to confirm the transcriptional identity of the TdT cells, which the authors succeed to do, it appears that n=1 and additional biological replicates would be recommended.

– From Figure 3G and 3K, it appears that cluster 5 can be subclustered in 2 subpopulations, have the authors attempted to perform further subclustering?

– The authors mention in the methods the redundancy in the TdT sequencing. Did this constitute a problem for the detection of the TdT transcripts in the single-cell analysis?

– No code for the single cell RNA-Seq analysis is provided and more information regarding QC (number of UMIs, % mitochondrial genes, etc) should be included.

– Many of the images have low resolution and are thus difficult to examine. In particular, Figure 1 would benefit from higher resolution images for instance at the lesion areas, so the reader can observe the labelling in individual cells.

– Figure 3—figure supplement 3 is completely missing.

– Figure 3—figure supplement 4 – can the authors show the expression of additional oligodendrocytes markers, such as PLP1, CNP, Mog, Mag, and transcription factors as Olig1 and Olig2, to further assess which type of transcripts are contaminating/being phagocytized by non-oligodendroglial cells.

– In Figure and Supplementary Figure legends, please state the number of ns corresponding to the representative figures.

– The Material and Methods part regarding animals should include much more detail and follow the ARRIVE guidelines as much as possible.

– In general, the manuscript is difficult to navigate, with much of the information in supplementary figures. We would recommend the authors reorganize the manuscript text and figures accordingly.

*Reviewer #3 (Recommendations for the authors):*

I think for Figure 1, I would like to see boxed regions of interest for the merged images shown below F and G at high power so we can decern at a minimum leptomeninges from parenchyma and overlapping from non-overlapping fluorescence (probably ventrolateral cord). I think the ventral brain/meningeal cluster would be better emphasized as a separate figure devoted to brain. Can you please include the n for each experiment shown and let us know how many times the experiment was replicated? In the methods, I did not see the methods for the flow experiments described in 1 C-E. Was there an analysis of myelin or is that planned for future experiments? I am not asking for this to be done but if you do have the data it would be nice to show in conjunction with the expression of CD74-TdT and B2m-TdT. Is the meningeal CD74-TdT expression in Figure 1 Supp 3F (baseline) more extensive than in Figure 1F (EAE score 0) – is this real or am I misreading the data? In Figure 1 supp 4, B2m-TdT expression seems extensive in the cortex (more than just layer 2) and in the corpus callosum. Are B2m-TdT and NeuN non-overlapping in the cortex (1 supp 4J)? If so, what lineage is B2m-TdT positive in 4J? What cell types are accounting for the intense B2m-TdT signal in the corpus callosum at baseline? What is the nature of the spindle-shaped cells that are Iba1 negative, B2m-TdT positive in the spinal cord (Figure 1 supp 4E)? What is the cell type in the CD74-TdT meningeal clusters in the brain (1I)?

The data in Figure 2 are central to the overall concept. Would like to see the HPF of the B2m-TdT glial culture with IFNγ (2A). The TdT signal in 2B seems highly polarized relative to olig2 – is this a consistent finding? For 2C, it would be helpful to see the olig2 (green) unmerged images. Does the quantitation of TdT+/Olig2+ cells in the EAE spinal cord (2D) refer to total spinal cord or lesions? Similarly, for 2H, is this quantification lesion-based or total cord? How were ROI's determined for the quantitative data in Figure 2 (Please include this in the methods)? For 2F please make it somehow clearer that the y-axis is PDGFR (-).

Why was the brain used for the scRNA seq in Figure 3 as opposed to the spinal cord? Was this analysis done on samples at a particular EAE stage? In the results text can you either cite or explain bias based on the efficiency of generating viable single cells for different cell populations? Similarly, would the immune-oligodendroglial cluster potentially be less viable since it is already considered stressed and thus the relative size underestimated? I may have missed this but how do the oligodendrocyte subclusters differ by CD74-TdT expression? The discussion on the highest MHC I expressing oligodendrocyte subcluster is very interesting (proteosome, PD-L1) and I would recommend expanding this discussion since this cluster is the focus of the paper. In the heat map (3K) I'm trying to understand what the individual vertical bars for a particular cluster and class II gene represent. Technical replicates? Experimental replicates? Perhaps p values are needed?

---

## [Author Response]

Essential revisions:1. The authors should explain the rationale for selecting B2m and Cd74, instead of other genes involved in these pathways.

We chose these MHC associated genes due to previous lines of evidence from in vitro OPC culture, in vivo mouse models, and human MS tissue that these transcripts are upregulated in oligodendroglia in response to inflammation, and because the degree of upregulation of transcript activity from minimal expression at baseline to much higher expression levels in the setting of inflammation was desirable for signal to noise. *B2m* was also chosen as a readout for both classical and non-classical MHC class I activity; *Cd74* is complementary, as the MHC class II invariant chain. We have added additional text (line 66) to indicate the rationale for choosing these genes for reporter generation.

2. MHC I has been traditionally described as expressed in all nucleated cells (https://www.ncbi.nlm.nih.gov/pmc/articles/PMC1783040/), although recent single-cell RNA-Seq studies suggest that this might not be the case. The data presented by the authors suggest that indeed activation of MHC I expression might be restricted, and it would be good if the authors discuss this aspect in relation to the current literature.

We appreciate the opportunity to expand on this concept. While all nucleated cells express MHC I, it has been known for some time that specific cells can increase MHC expression in response to viral infection or inflammatory stimuli especially IFNγ. A major impetus for making the MHC reporter mice was recent observations in human postmortem MS tissue and in EAE that subsets of glia, most notably oligodendrocyte lineage cells, had expression of MHC I, which may allow them to propagate CNS immune responses, but also then be targeted by CD8 T cells for cytotoxic cell death. This finding has implications for understanding why remyelination fails and why in some MS cases there is a paucity of OPCs. As with most advances in science, the observation is not entirely novel and there are data supporting enhanced MHC expression on oligodendrocyte lineage cells in response to sterile inflammation going back more than ten years, but the confluence of recent papers reporting this in MS tissues and the EAE model using single nucleus RNAseq has garnered widespread interest in the neuroscience community. We have added further discussion regarding this topic and the potential for neuronal MHC expression in the revised manuscript Line 332.

3. In Figure 1c, the difference observed in CD74-TdT is of much lower magnitude than B2m-TdT, could the authors comment on the reason why they think this is the case?

Figure 1C illustrates blood phenotyping for reporter animals sorted on all CD45-positive cells. Cd74 and MHC class II expression at baseline is only present in small populations of blood circulating immune cells. The flow histogram is indicating tdT expression in all CD45+ cells in the peripheral blood and there is a large population of tdT-negative cells in peripheral blood. Blood phenotyping was used in conjunction with PCR genotyping to identify reporter mice.

4. Line 162 – "B2m transcripts are upregulated in Cux2 expressing cortical layer II/II pyramidal neurons in human MS, which are susceptible to neurodegeneration3. Expression of MHC class I by these neurons may allow them to present antigen and render them more prone to death in inflammatory conditions." This is most likely referring to Supplementary Figure 4j, but no Cux2 staining to identify this cortical layer is provided. Also, it should be cortical layer II/III, not cortical layer II/II.

We attempted Cux2 immunostaining in response to this reviewer suggestion, but were unsuccessful with the commercial antibodies we tried. Therefore, we removed this comment from the manuscript.

5. The authors perform several quantifications of the lesion and non-lesion areas, but only mention DAPI hypercellularity to define a lesion, more details should be provided. Also, how many lesions were quantified per mouse, and which regions of the brain and spinal cord? Also, whenever the authors present a high-magnification view of lesions (e.g. Figure 2), it would be good to present a lower magnification image of the same region, to orient the reader.

We have provided further clarification in our methods about the regions of interest that were examined, the number of sections and regions of interest quantified per mouse (Methods, starting Line 529) and lesion analysis (Line 542). We have updated the figures to include representative examples of lower power images that were quantified for region of interest analysis. These panels are Figure 7B and Figure 7E for spinal cord and Figure 8C for brain. We have also added representative images of DAPI and CD45 hypercellular region compared to DAPI non-hypercellular region in Figure 7E. DAPI hypercellularity correlated with CD45 infiltrates in this additional staining.

6. While the authors present a supplementary video to confirm the colocalization of TdT and oligodendrocyte markers, it would be good to present Z-stacks of the images in Figure 2c. Also, the authors should describe in the methods the parameters by which they define which cells are double positive.

We provide Z-stack single images in Figure 8- source data 1. We have provided additional description about the quantification of Olig2+tdT+ cells in spinal cord and brain images in Methods, Line 529.

7. The authors find that the % of TdT+ Olig2+ cells out of Olig2+ cells was around 0.6% in the spinal cord, while Falcao and colleagues (Nature Medicine 2018) found this percentage to be 3-4%. The authors can discuss possible reasons for these differences.

In our study, the percentage of tdT+Olig2+ cells varied with the degree of inflammation and inflammatory model. In MOG35-55 EAE, the average %tdT+Olig2+ was 6% for *B2m^tdT^* and 0.6% for *Cd74^tdT^* in the spinal cord, and 6% for *B2m^tdT^* and 0.5% for *Cd74^tdT^* in the brain. For our EAE analysis we included animals that ranged from clinical score 1.5-3.5, which allowed our study to determine if higher clinical scores were correlated more MHC+ oligodendroglia, which is what we found on an individual animal level both with clinical scores and tdT+ intensity in ROIs as an overall readout of inflammatory activity. The percentage of MHC+ oligodendroglia ranged from 1-17% in *B2m^TdT^* and 0.1-1.4% in *Cd74^tdT^* EAE spinal cord. Models with high degrees of inflammatory infiltrate in the corpus callosum (AT-CUP) had higher percentages of tdT+Olig2+ cells, with a mean of 17% (range 3-34%) in *B2m^TdT^* and 1% (range 0.2-3.8%) in *Cd74^tdT^* AT-CUP brain. Thus, we found similar percentages to prior studies in animals with higher clinical scores and the degree of inflammatory activity on an individual animal basis correlated with MHC+ oligodendroglia across animals. We have also included brain regional breakdown, which revealed a higher percentage of MHC+ oligodendroglia in areas of higher inflammatory infiltrates in respective models: Figure 8—figure supplement 1B-D.

It is likely that the percentage of MHC+ oligodendroglia can differ when comparing quantification from different techniques, such as IHC staining and cells isolated for single-cell RNA sequencing. Tissue dissociation and steps involved in isolating cells from CNS tissue may introduce biases in quantification. When sorting tdT-positive cells from *CD74^tdT^* EAE brain, 3% of subclustered oligodendroglia were from the MHC class II *Cd74^tdT^* tdT+ sorted population, compared to a mean of 0.5% quantification from the EAE brain regional analysis; however, this varied on an individual animal basis and also by brain region. Quantification of MHC+ oligodendroglia previously published by Falcao et al. was based on mRNA transcript level and not protein or reporter expression, which could contribute to the differences in proportion of MHC expressing oligodendroglia.

8. In line 222 and Figure 2F, the authors indicate that the number of OPCs expressing TdT is much lower than CC1+ positive cells. It would be interesting to discuss why this might be the case and the possible implications of this difference.

We have added additional quantification of MHC+ OPCs based on PDGFRa expression to the analysis, as CC1 was not reliable in EAE tissue, showing high background and expression on some immune cells. We expanded the tissue quantified to include baseline, pre-symptomatic and symptomatic EAE tissue to determine if there is a loss of MHC+ OPCs with higher inflammatory activity and added this quantification to Figure 7J and Figure 7—figure supplement 1B-D. We found similar percentages of MHC+ OPCs across these conditions and did not find a correlation between the %PDGFRa+tdT+ cells and clinical score, overall %tdT+Olig2+ cells, or mean tdT intensity, suggesting there may not be a loss of MHC+ OPCs with higher inflammatory activity at the timepoints observed in this model. We have added additional discussion to the Results (starting on Line 167).

9. Figure 2I-J – The differences observed are borderline significant and very variable between lesions and non-lesions. Also in J, representative images are shown from different EAE scores.

The reviewer refers to lesion compared to non-lesion analysis. As we included animals with clinical score > 0 in our analysis, the degree of MHC+ oligodendroglia varied across animals and was positively correlated with higher clinical scores and the level of inflammatory activity. Expansion of this dataset allowed us to investigate differences in MHC expression in relation to EAE severity and inflammatory infiltrate. We provide additional description in the Methods (line 542) describing EAE lesion compared to non-lesion analysis and updated the figure to increase the n for lesion and non-lesion ROIs (Figure 7F). We have also included representative images from lesion compared to non-lesion areas in an individual animal in Figure7E.

10. Given that most of the EAE pathology occurs in the spinal cord, it is surprising that the authors present their single-cell studies with brain samples. In the Material and Methods, the spinal cord is also mentioned, with challenges in the sample preparation. The authors can integrate their dataset with other similar EAE datasets (Falcao et al., Nature Medicine 2018, Wheeler et al., Nature 2020) to compare the properties of the MHC1+ and/ or MHC2+ oligodendroglia (and astrocytes) they identify with the immune oligodendroglia and altered astrocytes in these datasets. Also, while the purpose of the single-cell experiment is to confirm the transcriptional identity of the TdT cells, which the authors succeed to do, it appears that n=1, and additional biological replicates would be recommended.

Unfortunately, we were not allowed to sort samples for over two hours using the core flow cytometer. Given this limitation, our spinal cord samples with higher myelin debris content were not sorted by the core facility, despite performing two rounds of myelin debris removal and optimizing our sample preparation. For our single cell RNA-seq experiments, 3 biological animal replicates were used for *B2m^tdT^* and 4 biological animal replicates were used for *Cd74^tdT^*. Animals were sacrificed at one time-point with clinical scores indicated in the Methods (Line 627). Biological replicates were pooled after flow cytometry for single cell sequencing.

In response to the reviewer comment regarding integration of previously published datasets, we performed additional analysis of our EAE brain oligodendroglial subclustering with Falcao et al. Nature Medicine 2018 EAE spinal cord oligodendroglial subclustering with methods described in line 703. We added an additional two figures Figure 11 and Figure 11—figure supplement 1 comparing our mouse EAE dataset with Falcao et al. mouse EAE dataset. Our dataset largely overlaps with the Falcao et al. dataset, with the exception of a paucity of OPCs – which we believe is due to enrichment of OPCs and lineage-traced OPCs with GFP reporter lines that were used in Falcao et al. dataset. Transcripts that were enriched in our subclusters also demonstrated enrichment in corresponding Falcao et al. clusters, based on location in UMAP of integrated datasets. Comparison of the two datasets offered insights into cluster 1, which was predominantly found in naïve EAE mouse in the Falcao et al. dataset and did not demonstrate enrichment of immunoproteasome, class II and class II transcripts, which we hypothesize represents a homeostatic OL population. Subclusters with higher MHC class I, MHC class II and cell stress transcripts were predominantly clustered separately from clusters enriched in naïve mice in Falcao et al. dataset and our cluster 1- homeostatic cluster.

11. From Figure 3G and 3K, it appears that cluster 5 can be subclustered in 2 subpopulations, have the authors attempted to perform further subclustering?

We indeed found that cluster 5 could be split into two clusters, which is noted in updated Figure 10 as 5.1 and 5.2. These clusters differed in terms of class II transcripts, with cluster 5.1 having relative enrichment compared to 5.2.

12. The authors mention in the methods the redundancy in the TdT sequencing. Did this constitute a problem for the detection of the TdT transcripts in the single-cell analysis?

tdT redundancy was a problem in sequencing of gel purified bands isolated from genomic DNA, but was not a problem for single cell sequencing. We were able to detect high levels of *tdT* transcripts in tdT+ sorted cells, as shown in Figure 9C, F.

13. No code for the single-cell RNA-Seq analysis is provided and more information regarding QC (number of UMIs, % mitochondrial genes, etc) should be included.

We have provided QC metrics from our dataset as a source data file (Figure 9-source data 2) and QC protocols and metrics were also added to the Methods (starting at Line 666). We have provided the code for our single cell analysis as a source data file (Figure 9- source data 3).

14. Many of the images have low resolution and are thus difficult to examine. In particular, Figure 1 would benefit of higher resolution images for instance at the lesion areas, so the reader can observe the labelling in individual cells.

We apologize for this issue, which occurred during formatting. We have updated the figures with higher resolution images.

15. Figure 3—figure supplement 3 is completely missing.

Figure 3 —figure supplement 3 was source data for our differential transcript analysis and is provided now as separate source data excel files for the corresponding figures.

16. Figure 3—figure supplement 4 – can the authors show the expression of additional oligodendrocytes markers, such as PLP1, CNP, Mog, Mag, and transcription factors as Olig1 and Olig2, to further assess which type of transcripts are contaminating/being phagocytized by non-oligodendroglial cells.

We have added expression of additional myelin, *Olig1*, and *Olig2* transcripts to Figure 10—figure supplement 1, as suggested.

17. In Figure and Supplementary Figure legends, please state the n for each experiment shown and how many times the experiment was replicated.

N values were added to figures and additional clarification is now indicated in the Methods. We have also provided all of the EAE IHC data used for quantification in additional source data of corresponding figures.

18. The Material and Methods part regarding animals should include much more detail and follow the ARRIVE guidelines as much as possible. Please clarify the methods for the flow experiments described in 1 C-E. Was there an analysis of myelin?

We have added additional information regarding the number of animals and experimental replicates for all experiments in the methods.

19. In general, the manuscript is difficult to navigate, with much of the information in supplementary figures. We would recommend the authors reorganize the manuscript text and figures accordingly.

We appreciate this comment and have reorganized the figures accordingly.

20. The presence of the 2A self-cleaving peptide should be highlighted when describing the reporter genes so that the reader can better understand reporter expression from the constructs.

We added a description of P2A function to Line 73 to help with describing this construct and the reporter expression.

21. The authors demonstrate the presence of the reporter fusion proteins on Western blot. Does the tomato portion need to be released from the fusion to provide fluorescence or is the presence of the fusion sufficient?

We are not able to determine the difference in fluorescence between fusion and cleavage *Cd74^tdT^* Cd74 protein without a method to differentiate these two protein isoforms in vitro or in vivo. We expect that based on high levels of tdT fluorescence found in *Cd74^tdT^* reporter animals and only 10-20% of Cd74 protein represented as fusion protein length on western blots that the presence of this fusion protein likely does not impact reporter expression and that 80-90% of the Cd74 is efficiently cleaved by P2A, which is sufficient for robust reporter expression. We have performed additional experiments assessing antigen presentation by *Cd74^TdT^* splenocytes to *OT-II* CD4 T cells in Figure 1—figure supplement 2. Notably, there was no difference between wild-type and *Cd74^tdT^* splenocytes in influencing CD4 T cell proliferation and activation, suggesting that the fusion protein does not impact Cd74 function, at least in terms of antigen presentation.

22. B2m and CD74 are necessary for cell surface expression of class I and II, respectively. Do the fusions influence (disrupt) cell surface class I or class II expression in reporter mice?

We did not detect fusion proteins in *B2m^tdT^* reporter animals. *Cd7t^TdT^* reporter animals did have 10-20% of Cd74 protein in spleen as a fusion protein. We think it is unlikely that the presence of a small amount of fusion protein influences MHC expression, as we saw robust reporter expression in cell types with known MHC class II and reporter expression co-localized with MHC class II (IA-IE) that was induced in the setting of inflammation.

23. Some comment on astrocyte expression, or lack thereof, of the reporters in the inflammatory models examined would be of interest.

We have added an additional figure investigating astrocyte expression of reporters in EAE in Figure 7—figure supplement 2. Astrocyte expression of both reporters, defined with *Sox9* to label astrocytes, was observed in EAE spinal cord tissue.

24. The mouse models report on class I and class II expression, which in turn report on the presence of inflammatory cytokines, particularly IFN-γ. Please comment on the spatial distribution of reporter-expressing cells in the CNS of their inflammatory models relative to the location of the immune cell infiltrates. This would provide some insight into the diffusion of the inflammatory cytokines away from the infiltrates.

This is an interesting question. We assessed regional and lesional differences in the location of reporter positive cells. For EAE spinal cord lesions there was a significant increase in *B2m^tdT^* positive oligodendroglia in lesions compared to non-lesions across the same animal (Figure 7F). For *Cd74^tdT^*, the higher clinical score animals demonstrated a trend for enrichment of tdT+ oligodendroglia, but this did not reach significance in the whole cohort, individual animal comparison. We also added analysis of whole spinal cord sections with CD45 percent area covered of each cord section compared to %tdT+Olig2+ cells for that section (Figure 7G). We observed a positive correlation between %tdT+Olig2+ cells and CD45 density in *B2m^tdT^* EAE spinal cord but observed no correlation in *Cd74^tdT^* tissue, in line with our lesion vs. non-lesion regional analysis.

For analysis of other inflammatory models such as AT-CUP, we expanded our analysis to include a regional breakdown of tdT+ oligodendroglia in the brain across inflammatory models (EAE, AT, CUP, AT-CUP) to determine if there was a regional enrichment in areas of inflammatory infiltrates (Figure 8I and Figure 8—figure supplement 1B-C). We observed an enrichment of tdT+ oligodendroglia in *B2m^tdT^* animals in areas with most prominent inflammatory infiltrates in each model; for cuprizone, the majority of tdT+ oligodendroglia were present in the corpus callosum, whereas in EAE the majority of tdT+ oligodendroglia were located in the ventral brain, notable for tdT+ meningeal clusters (Figure 5 A,B,D,E).

25. Figure 1, please show boxed regions of interest for the merged images shown below F and G at high power so we can decern at a minimum leptomeninges from parenchyma and overlapping from non-overlapping fluorescence (probably ventrolateral cord).

Thank you for this comment, we apologize for the poor resolution that occurred when importing figures. We have fixed this issue with a higher resolution version and included panels for cropped images in ventrolateral cord to illustrate Iba1/tdT staining for EAE spinal cord. In revised figures this is found in Figure 4 farthest panels on right.

26. The ventral brain/meningeal cluster would be better emphasized as a separate figure devoted to the brain.

We have provided an updated Figure 5 for the brain, demonstrating meningeal clusters for both reporters with animals of different scores and at baseline.

27. It would be nice to show it in conjunction with the expression of CD74-TdT and B2m-TdT. Is the meningeal CD74-TdT expression in Figure 1 Supp 3F (baseline) more extensive than in Figure 1F (EAE score 0)?

We have now included a baseline image comparing pre-clinical (EAE score 0) expression patterns in Figure 5. The meningeal expression was not notable at clinical score 0 animals in either reporter line.

28. In Figure 1 supp 4, B2m-TdT expression seems extensive in the cortex (more than just layer 2) and in the corpus callosum.

We agree with this comment. The more extensive expression appears to be neutrophil expression: aside from immunostaining for Iba1, vasculature, and NeuN, we did not observe any other cell types that could be accounting for this expression. When performing confocal imaging of these sections using these labeling methods, the remaining expression was diffuse, (see Figure 3C, L).

29. Are B2m-TdT and NeuN non-overlapping in the cortex (1 supp 4J)? If so, what lineage is B2m-TdT positive in 4J? What cell types are accounting for the intense B2m-TdT signal in the corpus callosum at baseline? What is the nature of the spindle-shaped cells that are Iba1 negative, B2m-TdT positive in the spinal cord (Figure 1 supp 4E)? What is the cell type in the CD74-TdT meningeal clusters in the brain (1I)?

B2m and NeuN colocalize in the cortex (Figure 3L). We attempted to perform Cux2 staining for this layer, but were unsuccessful with the commercial antibodies we tried. The spindle-shaped cells in Figure 3F striatum are endothelial cells/structure that are CD31 positive (Figure 3E). The corpus callosum meningeal cluster in Figure 5C had some microglia (TMEM119+Iba1+) and likely peripheral infiltrating macrophages that were TMEM119-Iba1+, (Figure 5G). In flow cytometry of *Cd74^tdT^* EAE, brain infiltrating Clec12a+ infiltrating myeloid cells and microglia both exhibited tdT+ expression, with infiltrating myeloid cells having higher tdT expression compared to microglia (Figure 6B).

30. In Figure 2, please provide the HPF of the B2m-TdT glial culture with IFNγ (2A). The TdT signal in 2B seems highly polarized relative to olig2 – is this a consistent finding? For 2C, it would be helpful to see the olig2 (green) unmerged images. Does the quantitation of TdT+/Olig2+ cells in the EAE spinal cord (2D) refer to total spinal cord or lesions? Similarly, for 2H, is this quantification lesion-based or total cord? How were ROI's determined for the quantitative data in Figure 2 (Please include this in the methods)? For 2F please make it somehow clearer that the y-axis is PDGFR (-).

We have provided high power images of a *B2m^tdT^* glial culture in Figure 1I. It does appear that some OPCs have areas of higher tdT expression opposing Olig2, although this did not seem consistent across all OPCs in culture. For EAE confocal imaging, we have added Olig2 unmerged images (Figure 7A far right panel). Quantification of EAE spinal cord refers to total ROI mean with ROIs now depicted in Figure 7C. A detailed description of ROI quantification and number of ROIs has been added to the Methods (Line 529). We have also provided a spreadsheet of quantification of all ROIs, sections and animals for EAE and other inflammatory models analyzed in Figure 7-source data 1 and Figure 8- source data 2. Similarly, 2H, now Figure 7H is total ROI mean. For lesional analysis, we categorized images at lesion or non-lesion based on DAPI hypercellularity for 8-9 reporter animals and compared lesion to non-lesion ROIs across individual animals for Figure 7F. We also added an additional analysis of a small cohort of higher scoring EAE animals (3 animals per reporter) and performed CD45 staining and quantified CD45 intensity across the whole spinal cord section compared to whole section %tdT+Olig2+ cells (Figure 7G and Figure 7—figure supplement 1B-C). On a sectional basis, we saw a significant negative correlation between the degree of CD45 intensity and percentage of *Cd74^tdT^* tdT+ oligodendroglia, suggesting a possible loss of MHC II+ oligodendroglia in areas of higher inflammatory activity. We adjusted the graph for PDGFRa quantification to reflect PDGFRa+ cells (Figure 7J) and added additional animals to the quantification.

31. Why was the brain used for the scRNA seq in Figure 3 as opposed to the spinal cord? Was this analysis done on samples at a particular EAE stage? In the results text can you either cite or explain bias based on the efficiency of generating viable single cells for different cell populations? Similarly, would the immune-oligodendroglial cluster potentially be less viable since it is already considered stressed and thus the relative size underestimated? How do the oligodendrocyte subclusters differ by CD74-TdT expression?

Due to technical limitations, we used the brain instead of the spinal cord for single cell sequencing. Both brain and spinal cord were dissociated, however the flow cytometry core at JHU refused to sort the spinal cord samples, as the percentage of myelin debris was too high compared to percentage of cells. Despite our attempts to reduce myelin debris with multiple rounds of debris removal kit in optimization experiments prior to sort, we did not reach a level that passed this QC step. For the *Cd74^tdT^* experiment, we pooled four animals at 14 days post-immunization with clinical scores of 2.5, 3.,3.5, 3.5. For the *B2m^tdT^* experiment, we pooled three animals 13 days post-immunization with clinical scores of 2.5, 4, 4. We used papain for dissociation, as we found this was the best for isolating higher percentages of CNS resident cells and this may not allow for isolation of certain cell types such as astrocytes, we added this to text Line 240. It is possible tdT+ oligodendrocytes were underestimated in our cells isolated by sequencing, however in our *Cd74^tdT^* samples that we performed oligodendroglia subclustering the percentage of tdT+ sorted oligodendroglia was 3% compared to tdT- sorted oligodendroglia and this percentage was relatively comparable to percentages found in *Cd74^tdT^* EAE brain by IHC, Figure 8—figure supplement 1C. *Cd74^tdT^* tdT-positive sorted oligodendroglia were found almost exclusively in cluster 5. We have provided an additional panel to illustrate *Cd74* transcript compared *Cd74^tdT^* tdT-positive sorted cells in Figure 10F.

32. The discussion on the highest MHC I expressing oligodendrocyte subcluster is very interesting (proteosome, PD-L1) and I would recommend expanding this discussion since this cluster is the focus of the paper.

We added additional discussion about this point (line 314). It is intriguing that MHC class I oligodendroglia are present in high numbers in some of our inflammatory models. We observed that *PD-L1* transcript levels were highest in the highest MHC class I transcript clusters. This may suggest MHC class I oligodendroglia are protected from CD8 mediated cytotoxicity.

33. In the heat map (3K) I'm trying to understand what the individual vertical bars for a particular cluster and class II gene represent? Technical replicates? Experimental replicates? Perhaps p values are needed?

Vertical bars represent each oligodendroglial subcluster from Figure 10A and are colored accordingly in the bar. Each vertical line represents individual cells from the clusters. We added violin plots of MHC class II transcript expression in Figure 10G, for additional illustration of these data.